# A Survey of Reasoning and Agentic Systems in Time Series with Large Language Models

**Ching Chang**[§][¶][*]                    *chingchang0730@ucla.edu*

**Yidan Shi**[§]                          *yidanshi@g.ucla.edu*

**Defu Cao**[†]                           *defucao@usc.edu*

**Wei Yang**[†]                           *wyang930@usc.edu*

**Jeehyun Hwang**[§]                       *jeehyunhwang@cs.ucla.edu*

**Haixin Wang**[§]                         *whx@cs.ucla.edu*

**Jiacheng Pang**[†]                       *pangj@usc.edu*

**Wei Wang**[§]                            *weiwang@cs.ucla.edu*

**Yan Liu**[†]                             *yanliu.cs@usc.edu*

**Wen-Chih Peng**[¶]                       *wcpeng@cs.nycu.edu.tw*

**Tien-Fu Chen**[¶]                        *tfchen@cs.nycu.edu.tw*

[§] *University of California, Los Angeles*
[†] *University of Southern California*
[¶] *National Yang Ming Chiao Tung University*

**Reviewed on OpenReview:** *https://openreview.net/forum?id=l3QW42g6u3*

## Abstract

Time series reasoning treats time as a first-class axis and incorporates intermediate evidence directly into the answer. This survey defines the problem and organizes the literature by reasoning topology with three families: direct reasoning in one step, linear chain reasoning with explicit intermediates, and branch-structured reasoning that explores, revises, and aggregates. The topology is crossed with the main objectives of the field, including traditional time series analysis, explanation and understanding, causal inference and decision making, and time series generation, while a compact tag set spans these axes and captures decomposition and verification, ensembling, tool use, knowledge access, multimodality, agent loops, and LLM adaptation regimes. Methods and systems are reviewed across domains, showing what each topology enables and where it breaks down in faithfulness or robustness, along with curated datasets, benchmarks, and resources that support study and deployment (with an accompanying repository at https://github.com/blacksnail789521/Time-Series-Reasoning-Survey). Evaluation practices that keep evidence visible and temporally aligned are highlighted, and guidance is distilled on matching topology to uncertainty, grounding with observable artifacts, planning for shift and streaming, and treating cost and latency as design budgets. We emphasize that reasoning structures must balance capacity for grounding and self-correction against computational cost and reproducibility, while future progress will likely depend on benchmarks that tie reasoning quality to utility and on closed-loop testbeds that trade off cost and risk under shift-aware, streaming, and long-horizon settings. Taken together, these directions mark a shift from narrow accuracy toward reliability at scale, enabling systems that not only

---

[*]Corresponding author

analyze but also understand, explain, and act on dynamic worlds with traceable evidence and credible outcomes.

## Contents

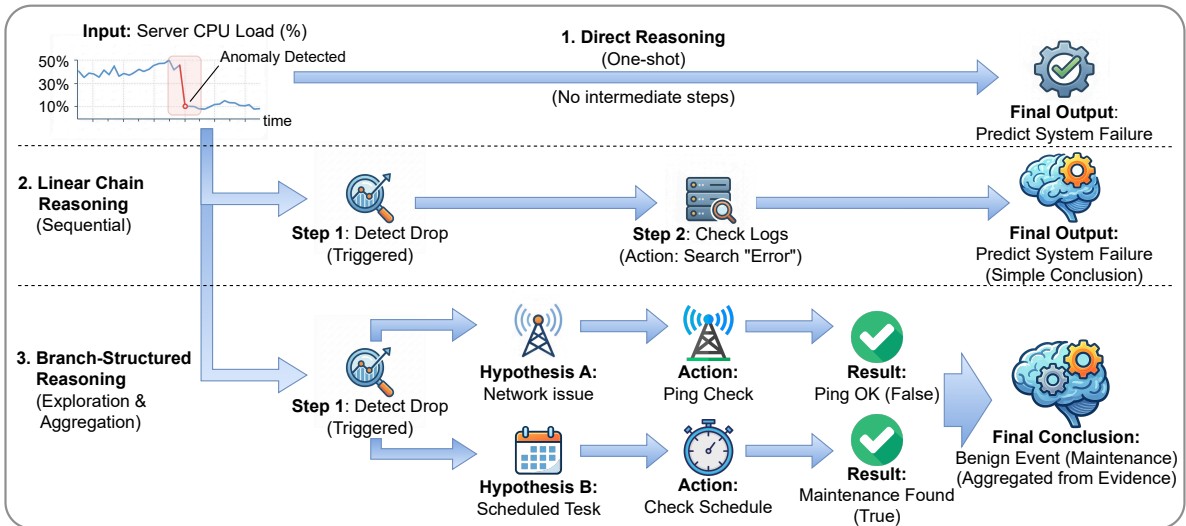

Figure 1: Illustrative reasoning traces on an anomaly detection task. (1) Direct Reasoning: The model maps the input series directly to a prediction (e.g., system failure) in one step, potentially missing context. (2) Linear Chain Reasoning: A sequential process detects the drop, retrieves logs, and then predicts, adding interpretability but remaining on a single track. (3) Branch-Structured Reasoning: An exploratory process where the model spawns multiple hypotheses (e.g., network issue vs. scheduled task), validates them against external tools in parallel, and aggregates the evidence to conclude the event is benign. This illustrates how complex topology enables self-correction.

# 1 Introduction

Time series data are common in everyday life, recording how variables evolve and interact over time in fields like finance, healthcare, energy, climate, transport, and manufacturing processes (Chang et al., 2025c; Ou et al., 2024; Chang et al., 2025b; 2024c; Zhao et al., 2020; Cao et al., 2024b; Niu et al., 2024; Cao et al., 2022; Li et al., 2025c). Decades of effort have made time series analysis one of the key methodologies used in monitoring, forecasting, diagnostics, and decision-making, and it has countless uses in areas such as risk modeling and patient monitoring, demand, and predictive maintenance (Chang et al., 2024a; Lin et al., 2024; Chang et al., 2024b; Lo et al., 2024; Cao et al., 2020). Existing surveys on time series have generally focused on modeling and algorithmic methods. These comprise surveys on deep learning forecasting methods (Lim & Zohren, 2021; Torres et al., 2021; Mahalakshmi et al., 2016; Liu et al., 2021; Benidis et al., 2022), architectures employing transformers (Wen et al., 2023), anomaly detection (Zamanzadeh Darban et al., 2024), classification (Ismail Fawaz et al., 2019), clustering (Liao, 2005), discovery of motifs (Torkamani & Lohweg, 2017), change-point detection (Aminikhanghahi & Cook, 2017), segmentation (Keogh et al., 2004), compression (Chiarot & Silvestri, 2023), and data augmentation (Victor & Ali, 2024; Wen et al., 2021). Respectively, these works are concerned with increasing predictive accuracy, representation, and efficiency in dealing with sequential temporal data.

However, many emerging applications demand more than prediction. Domains such as personalized healthcare, adaptive risk management, and autonomous systems require models that can explain their outputs, reason about counterfactuals, and decide among alternative actions. These demands underscore that advancing time series analysis requires structured and reliable reasoning. Despite this breadth, the literature to date has not covered reasoning, explanation, or agent-based decision-making under time series. To our best knowledge, no work has been devoted to investigating how methods under time series can be used toward enabling higher-level reasoning or policy-oriented actions.

The advent of large language models (LLMs) is another turning point. Besides fitting patterns, LLMs can exhibit step-by-step reasoning traces (Ke et al., 2025; Zhang et al., 2024c; Huang & Chang, 2023; Chu et al.,

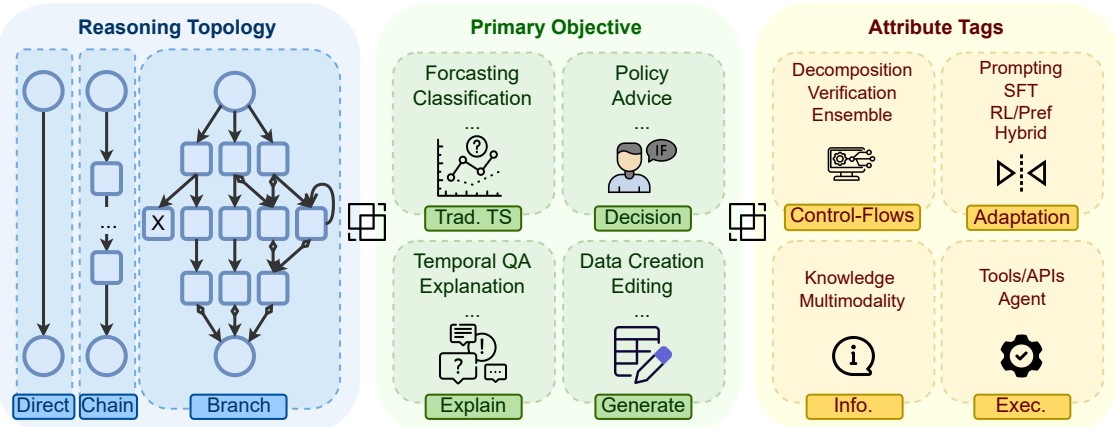

Figure 2: Framework of reasoning topologies and primary objectives, complemented by lightweight attribute tags.

2024; Zhang et al., 2025g; Yang & Thomason, 2025; Xiao et al., 2024), articulate causal hypotheses (Li et al., 2025d; Liu et al., 2025f; Kiciman et al., 2023; Zhang et al., 2022; Cao et al., 2023), and interact with external tools and environments (Shen, 2024; Ferrag et al., 2025; Yang et al., 2025a; Chen et al., 2025b). When incorporated into agentic systems, they gain the capacity for planning (Huang et al., 2024; Wei et al., 2025), reflection (Renze & Guven, 2024; Ji et al., 2023), and continual adaptation (Fujii et al., 2024; Shi et al., 2025), changing time series modeling from static prediction to interactive and explanatory processes(Ye et al., 2024). This shift opens up the space of downstream tasks: instead of just prediction or anomaly detection, models are now expected to handle causal analysis, natural language reasoning, simulating and editing temporal signals, and making policy-driven decisions.

Building on this transformation, our survey is structured on the basis of three intersecting trends that shape the future landscape. First, time series data are increasingly widespread and significant, driving practical systems that require clarity, versatility, and strong decision-making under uncertainty. Second, LLMs and multimodal LLMs have demonstrated unprecedented flexibility in reasoning and generalization, creating opportunities to recast time series problems in natural language and symbolic forms. Third, the rise of autonomous agents driven by LLMs allows models not only to analyze time series but also to act upon them—through simulation, intervention, or iterative decision loops.

Motivated by these developments, we define and study time series reasoning (TSR) as the class of methods where LLMs explicitly execute structured reasoning procedures over temporally indexed data, potentially enriched by multimodal context and agentic systems. Figure 1 illustrates this evolution on a concrete anomaly detection task. The figure also previews a recurring analytical theme of the survey: topology changes what evidence is visible, where errors can be checked, and whether competing hypotheses can be revised or aggregated. Direct reasoning offers speed but exposes little intermediate evidence; linear chain reasoning adds sequential verification while remaining vulnerable to error propagation; and branch-structured reasoning supports exploration across hypotheses (e.g., network error vs. maintenance) at the cost of higher orchestration and reproducibility burden.

This survey presents the first systematic taxonomy of the field, organized around distinct reasoning topologies and primary objectives, and complemented by lightweight attribute tags that capture control-flow operators (such as decomposition, verification, and ensembling), actors (including tool use and agentic loops), modality and knowledge access, and adaptation regimes specific to LLMs, as illustrated in Figure 2.

This survey makes three contributions. (i) We introduce the first systematic taxonomy of time series reasoning, structured along two complementary axes: reasoning topologies (execution structures) and primary objectives (task intents), and further enriched with lightweight attribute tags that capture control-flow operators, actors

(tools and agent loops), modality and knowledge access, and LLM adaptation regimes. (ii) We provide an integrated review that not only analyzes patterns across reasoning topologies and objectives in research papers, but also categorizes complementary contributions such as datasets, benchmarks, surveys, tutorials, position and vision papers—highlighting how these works support and shape the development of time series reasoning. (iii) We highlight open problems in evaluation and benchmarking, multimodal fusion and cross-modal alignment, retrieval and knowledge grounding, long-context reasoning, memory and efficiency, agentic control and tool use, as well as causal inference and decision support—laying out a research agenda for the next stage of time series reasoning.

The rest of the paper is organized as follows. Section 2 formalizes the notion of time series reasoning and introduces our taxonomy, decision sketches, and a systematic labeling pipeline that annotates each paper with reasoning topology, primary objectives, and attribute tags. Sections 3–5 analyze the three reasoning topologies in depth, while Section 6 surveys datasets, benchmarks, evaluation protocols, and auxiliary resources, including recent controversies and counter-evidence (§6.3) that highlight limitations and ongoing debates. Finally, Section 7 outlines open problems and future directions. Together, these sections establish a unified taxonomy, a reproducible labeling of over one hundred papers, and a synthesis of methodological trends and challenges, aiming to serve both researchers developing novel reasoning systems for time series and practitioners seeking a structured guide to the current landscape and its open questions.

## 2 Background and Taxonomy

### 2.1 What We Mean by Time Series Reasoning

Time series reasoning (TSR) refers to methods that operate over temporally indexed data while executing an explicit reasoning procedure. These methods are increasingly enabled by large language models (LLMs) and multimodal LLMs, which can articulate reasoning traces, interact with external tools, and operate as autonomous agents. In doing so, they not only strengthen traditional time series analysis tasks such as forecasting, anomaly detection, and classification, but also extend the scope of what is possible by enabling explanation, intervention, and generation of temporal dynamics. Such reasoning may take the form of single-step inference, multi-step decomposition, or branching exploration that allows both divergence and feedback across reasoning paths, reflecting an expanded view of how models can reason with time series. We therefore distinguish TSR from generic LLM-for-time-series applications by requiring that temporal data shape an explicit reasoning procedure, evidence trace, tool interaction, or agentic workflow, rather than merely serving as input to a black-box predictor.

Formally, we define the problem setting as learning a parameterized model $\mathcal{M}_\theta$ (an agent, LLM, or hybrid architecture) that maps a composite input to an output $Y$. The input consists of the historical time series $\mathbf{X}_{1:T} = (\mathbf{x}_1, \ldots, \mathbf{x}_T)$ where $\mathbf{x}_t \in \mathbb{R}^D$, a textual context or instruction $C$ (e.g., "Predict the next 24 steps" or "Why did the anomaly occur?"), and accessible auxiliary knowledge $K$ (e.g., retrieved documents, API tools, or static facts). The core distinction of TSR lies in how $\mathcal{M}_\theta$ utilizes an intermediate reasoning trace $\mathcal{Z}$ to bridge the inputs $(\mathbf{X}_{1:T}, C, K)$ and the final output $Y$.

In our taxonomy, TSR is defined by three complementary components: the **Reasoning Topology** (Section 2.3), which specifies the execution structure of $\mathcal{Z}$; the **Primary Objective** (Section 2.4), which clarifies the main intent (loss function) of the reasoning process; and a set of **Attribute Tags** (Section 2.5), which describe auxiliary properties such as control-flow, actors, modality, and adaptation. The first two levels—reasoning topology and primary objective—are *mutually exclusive*: each paper is assigned exactly one topology and exactly one objective, according to the minimal structural test and the dominant evaluation focus. In contrast, attribute tags are *non-exclusive*: a paper may carry multiple tags at once, since it can simultaneously employ decomposition, use tools, access multimodal inputs, and involve specific adaptation regimes. Figure 3 presents the taxonomy with reasoning topologies and primary objectives. For clarity, attribute tags are not included in this figure and are discussed separately in Section 2.5.

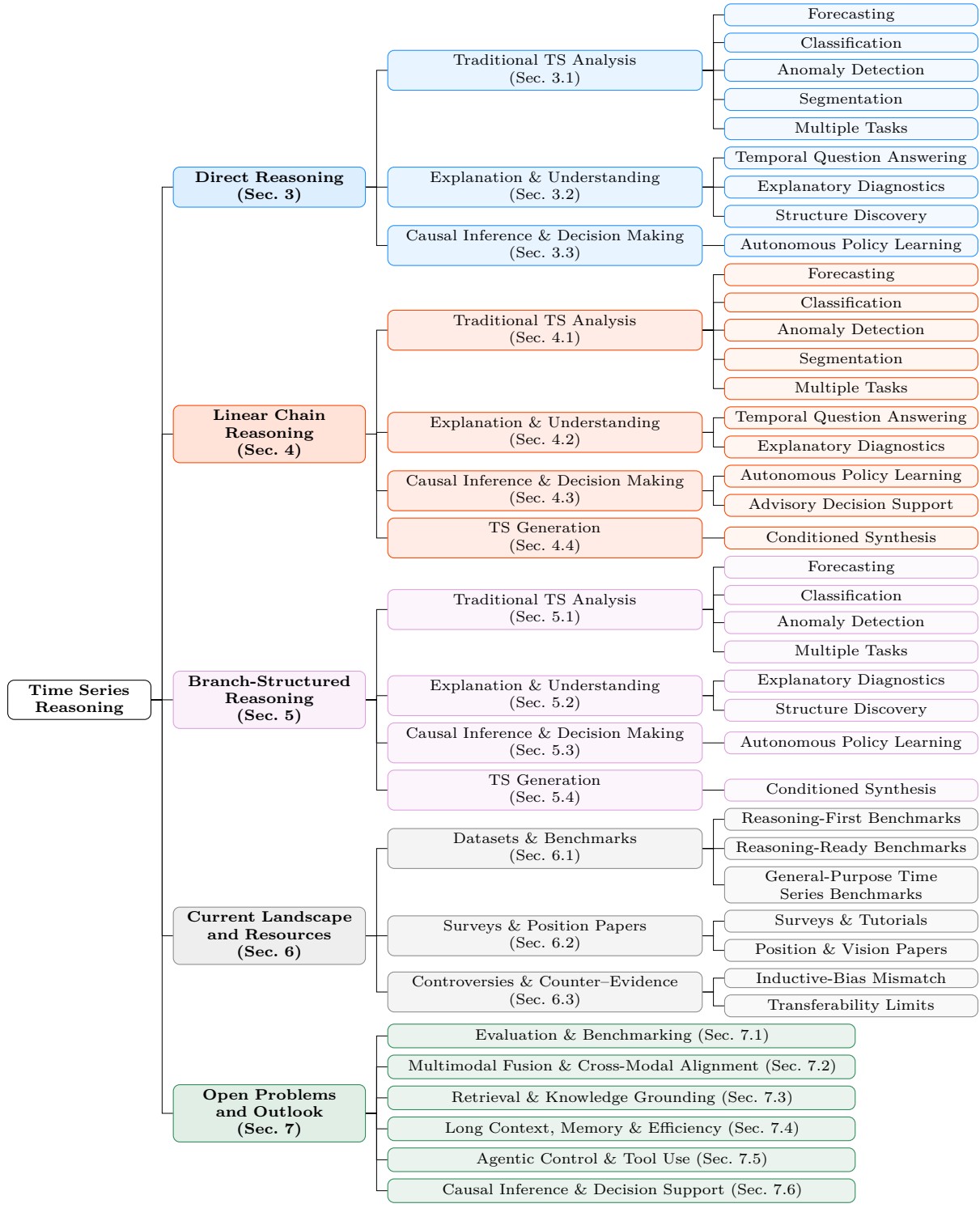

Figure 3: Taxonomy of time series reasoning literature

## 2.2 Survey Corpus Construction

We constructed the survey corpus as an expert-curated collection of work available up to April 2026. Papers were gathered through keyword searches, source and venue scanning, arXiv and OpenReview searches, and

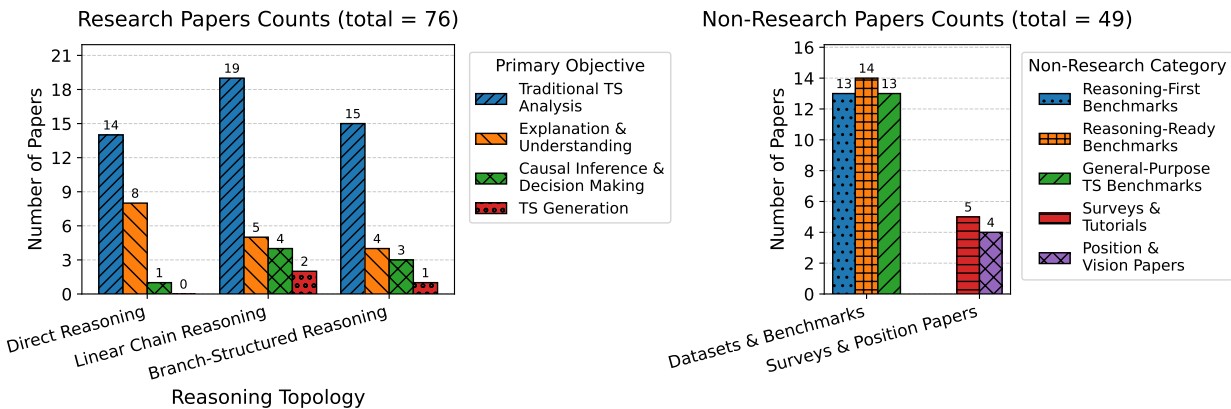

Figure 4: Number of surveyed papers: research (left) and non-research (right).

follow-up through references and closely related work. The search themes covered combinations of terms around time series reasoning, LLMs for time series, time series forecasting, anomaly detection, temporal question answering, multimodal time series, tool use, agents, and decision support. We also considered adjacent benchmark, survey, tutorial, position, and vision papers when they helped clarify the resources, controversies, or evaluation practices surrounding time series reasoning.

We included a paper when time series data were central to the problem setting and the method, benchmark, or analysis involved LLMs, multimodal LLMs, explicit reasoning traces, tool use, agentic workflows, or other reasoning-oriented mechanisms. We excluded conventional time series papers without an LLM or reasoning component, generic LLM papers without a substantive temporal-data setting, duplicate versions of the same work, and papers where time series appeared only incidentally. When multiple versions of a work were available, we used the conference version or the latest complete version when possible. The resulting corpus is summarized in Figure 4, while the full labeled list is provided in Appendix A.

For each included research paper, labels were assigned from the method description and reported execution flow. The reasoning topology is determined by the observable structure of the reasoning trace: no explicit intermediate reasoning yields direct reasoning; a single ordered sequence yields linear chain reasoning; and alternatives, aggregation, feedback, revision, debate, or search within a run yield branch-structured reasoning. The primary objective is assigned according to the dominant evaluation target or main claimed contribution, while task labels refine the concrete problem setting. Attribute tags are assigned only when the corresponding behavior is explicitly evidenced, and they remain non-exclusive so that secondary behaviors in hybrid systems can be recorded without changing the primary topology or objective. These labels describe execution structure and survey organization; they do not by themselves certify correctness, faithfulness, causal validity, or robustness, which must be evaluated separately.

## 2.3 Reasoning Topology

We identify three mutually exclusive reasoning topologies, corresponding to minimal structural tests over the reasoning trace $\mathcal{Z}$: direct reasoning, linear chain reasoning, and branch-structured reasoning (as illustrated in Figure 5). These topologies form a spectrum of increasing complexity: from direct reasoning with a single step, to linear chains of sequential steps, to branch-structured reasoning that supports in-trajectory exploration, reconnection, and feedback. This progression highlights how reasoning can evolve from simple one-shot inference to richer temporal analysis that coordinates among alternative paths.

**Direct Reasoning.** Direct reasoning denotes the simplest form of execution: a single-step inference or tool call without any intermediate reasoning traces (i.e., $\mathcal{Z} = \emptyset$). Mathematically, the model approximates the conditional distribution directly:

$$P(Y \mid \mathbf{X}_{1:T}, C, K; \theta) \tag{1}$$

The model jumps directly from input to output, producing a forecast, classification, or anomaly label without decomposing the problem or iterating over solutions. Such execution may be implicit, with internal reasoning hidden within the model's parameters, or minimal, with only the final output exposed. Direct reasoning is commonly used as a baseline or reference point, since it maximizes efficiency and requires no orchestration overhead, but it also limits interpretability, robustness to errors, and adaptability to complex or multi-stage tasks. Despite these limitations, direct reasoning remains prevalent in practice for straightforward forecasting benchmarks, anomaly detection pipelines, or descriptive question answering when transparency and intermediate supervision are not required.

**Linear Chain Reasoning.** Linear chain reasoning extends beyond direct inference by introducing a sequence of reasoning steps $\mathcal{Z} = (z_1, z_2, \ldots, z_k)$ arranged in a straight path. Each step depends on the output of the previous one, forming a logical progression such as step-by-step forecasting, causal analysis, or explanation. Formally, the generation process factors as:

$$P(Y, \mathcal{Z} \mid \mathbf{X}_{1:T}, C, K; \theta) = \underbrace{P(Y \mid \mathcal{Z}, \mathbf{X}_{1:T}, C, K; \theta)}_{\text{Answer Generation}} \prod_{i=1}^{k} \underbrace{P(z_i \mid z_{<i}, \mathbf{X}_{1:T}, C, K; \theta)}_{\text{Reasoning Step}} \tag{2}$$

This sequential structure allows intermediate states to be explicitly represented, inspected, or revised in later steps, thereby offering greater interpretability and modularity than direct reasoning. Crucially, the dependence on $K$ at each step allows for dynamic retrieval (e.g., tool use) where step $z_i$ retrieves information required for $z_{i+1}$. Chains are especially useful when tasks naturally unfold in stages, or when human users or downstream systems benefit from observing intermediate results. However, the linear chain topology remains restricted to a single path with no branching, feedback loops, or cross-branch aggregation, which limits its flexibility in exploring multiple hypotheses or adapting dynamically during execution.

**Branch-Structured Reasoning.** Branch-structured reasoning represents any topology where the reasoning trace $\mathcal{Z}$ forms a directed graph (e.g., a tree or DAG), allowing the model to explore multiple paths within a single execution. Branches may arise when the model explores different hypotheses, candidate forecasts, explanations, or plans in parallel or sequentially, creating sibling nodes that diverge from a common ancestor. The final output is typically derived via an aggregation function $\psi$ (e.g., majority vote, ranking, or fusion) over the set of terminal reasoning states $\{z_{\text{leaf}}^{(j)}\}$:

$$Y = \psi\left(\{z_{\text{leaf}}^{(j)}\}_{j=1}^{M}, \mathbf{X}_{1:T}, K; \theta\right) \tag{3}$$

Branching may involve simple divergence, where alternative paths are explored independently, or more complex interactions, where later steps feed back to earlier ones or combine information from multiple branches. For example, feedback loops can revise or regenerate earlier outputs, while cross-branch operations can create new steps that depend on several existing paths. Aggregation is also part of this class: a parent node may select or rank among its children, or a new step may fuse multiple branches into a shared outcome. Compared to linear chain reasoning, branch-structured reasoning enables exploration of alternatives, adaptive revision of earlier steps, and reuse of intermediate results, but it also raises challenges such as controlling the growth of branches, handling feedback cycles, defining stopping conditions, and ensuring reproducibility. To manage combinatorial growth, branch-structured methods typically employ pruning, which eliminates low-promise branches early (for example, beam-style cutoffs or budgeted search) within the same execution trace.

## 2.4 Primary Objective

The primary objective of a TSR method captures the ultimate purpose of its reasoning process. While reasoning topology specifies *how* a model arrives at an answer, the primary objective defines the intended outcome of that process. This distinction is essential, since structurally similar reasoning strategies may pursue very different ends—for example, a chain of reasoning might be used either to forecast future values, to explain causal mechanisms, or to simulate new scenarios. We group objectives into four broad categories that span the major goals of time series reasoning: traditional time series analysis, explanation and understanding,

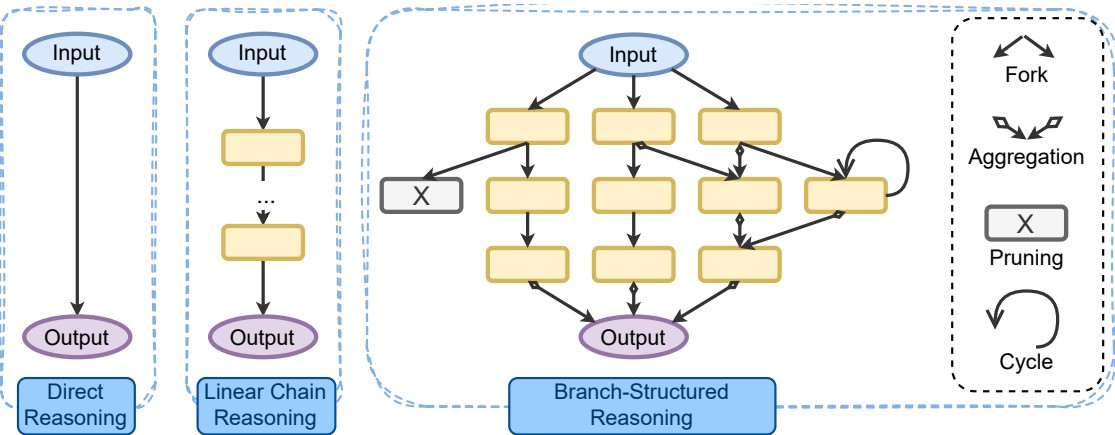

Figure 5: Three types of reasoning topologies: direct reasoning, linear chain reasoning, and branch-structured reasoning. Yellow boxes represent intermediate reasoning steps. Branch-structured reasoning additionally supports four structures: fork, aggregation, pruning, and cycle.

causal inference and decision making, and time series generation. These categories provide a complementary view to reasoning topology, helping us compare methods not only by how they reason, but also by why they reason.

### 2.4.1 Traditional Time Series Analysis

This category covers predictive and descriptive tasks that directly model temporal dynamics. It serves as the foundation of time series reasoning, focusing on core supervised objectives such as predicting future values, assigning labels, detecting irregularities, and segmenting sequences into meaningful parts. Mathematically, these tasks share a common goal of minimizing a predictive loss $\mathcal{L}_{\text{pred}}$ between the model output $f_\theta(\cdot)$ and a ground truth target $\mathbf{Y}_{\text{true}}$ (which may be future values $\mathbf{X}_{T+1:H}$, class labels, or anomaly masks), conditioned on the reasoning trace:

$$\mathcal{L}_{\text{pred}}(\theta) = \mathbb{E}\left[\ell(f_\theta(\mathbf{X}_{1:T}, \mathcal{Z}, C, K), \mathbf{Y}_{\text{true}})\right] \tag{4}$$

where $\ell$ is a task-specific metric (e.g., MSE for forecasting, Cross-Entropy for classification).

**Forecasting.** Reasoning-oriented forecasting treats prediction not only as extrapolation of past values but as an explicit reasoning process that interprets temporal patterns and conditions on context before projecting into the future. Evaluation emphasizes point accuracy and probabilistic calibration, typically using mean absolute error, root mean squared error, and the continuous ranked probability score.

**Classification.** Reasoning for classification involves mapping temporal sequences to semantic categories through structured prompts, cross-modal alignment, or stepwise inference, rather than treating label assignment as a black-box mapping. Evaluation focuses on robustness under imbalance and overall correctness, commonly using accuracy, F1, area under the receiver operating characteristic curve, and area under the precision–recall curve.

**Anomaly Detection.** Here reasoning is used to discern whether irregular points or intervals are true anomalies, often by contrasting candidate explanations, incorporating domain knowledge, or verifying suspicious patterns against context. Evaluation prioritizes correct localization and event quality, commonly using precision, recall, F1, and event-level F1, sometimes alongside detection delay.

**Segmentation.** Reasoning-based segmentation decomposes a sequence into meaningful sub-intervals or detects change points by combining statistical cues with interpretable decision rules, producing boundaries

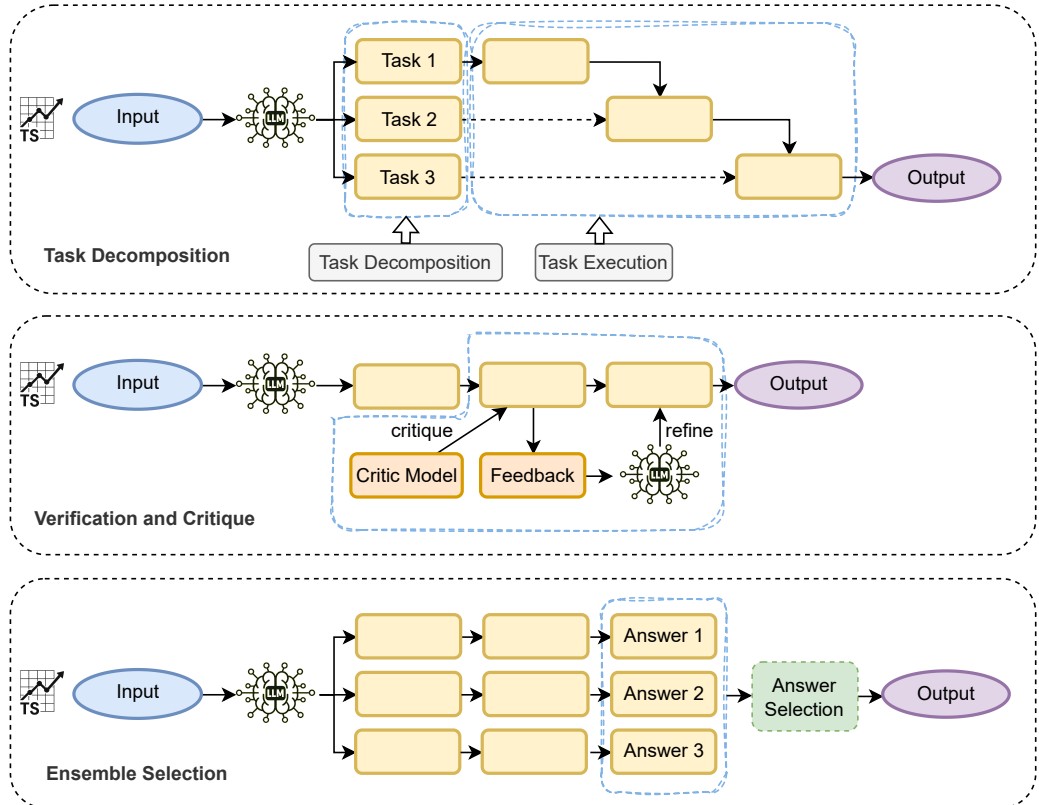

Figure 6: Control-flow operators: task decomposition, verification and critique, and ensemble selection.

that reflect underlying dynamics. Evaluation emphasizes boundary accuracy and stability, for example boundary F1 and mean absolute boundary error.

**Multiple Tasks.** Unified reasoning frameworks tackle several objectives simultaneously, for example forecasting and classification, by reusing reasoning traces or branching workflows across tasks. Evaluation reports task-specific metrics for each included objective, for example mean squared error for forecasting and F1 for classification.

### 2.4.2 Explanation and Understanding

This category emphasizes reasoning that produces human-interpretable insights about temporal phenomena rather than raw predictions. It encompasses objectives such as answering scoped temporal questions, generating diagnostic narratives that clarify underlying causes, and discovering structural representations like causal tuples or symbolic rules. The objective is generally to maximize the likelihood of a textual explanation or structured answer $Y = (w_1, \ldots, w_N)$ that is faithful to the input signals and reasoning trace:

$$\mathcal{L}_{\text{gen}}(\theta) = -\sum_{j=1}^{N} \log P(w_j \mid w_{<j}, \mathbf{X}_{1:T}, \mathcal{Z}, C, K; \theta) \tag{5}$$

**Temporal Question Answering.** Reasoning appears as the ability to parse a question about a time-indexed signal, retrieve relevant evidence, and articulate a direct answer grounded in temporal context. Evaluation measures answer correctness and grounding quality, commonly using question answering accuracy, exact match, and faithfulness or sufficiency scores.

**Explanatory Diagnostics.** These methods emphasize reasoning that connect observed outcomes to underlying causes, producing diagnostic narratives or structured explanations that clarify temporal behavior. Evaluation centers on explanation quality, commonly using human- or model-rated helpfulness, faithfulness, and coverage of salient events.

**Structure Discovery.** Reasoning is made explicit by generating candidate causal tuples, symbolic rules, or mechanistic abstractions and refining them into explanatory structures that capture time-series dependencies. Evaluation focuses on structure recovery quality, for example structural Hamming distance, edge precision and recall, and rule fidelity or coverage.

### 2.4.3 Causal Inference and Decision Making

This category focuses on reasoning about interventions and their outcomes in temporal settings. It covers autonomous policy learning, where models derive and execute action strategies directly from temporal states, as well as advisory decision support, where systems provide justified recommendations or what-if analyses to assist human decision makers. These tasks can be formalized as maximizing the expected utility or return $R$ of a policy $\pi$ derived from the model's reasoning:

$$J(\pi_\theta) = \mathbb{E}_{\tau \sim \pi_\theta} \left[ \sum_{t=0}^{H} \gamma^t R(\mathbf{x}_t, a_t) \right], \quad \text{where } a_t \sim \pi_\theta(\cdot \mid \mathbf{X}_{1:t}, \mathcal{Z}_t, C, K) \tag{6}$$

**Autonomous Policy Learning.** Reasoning traces in this setting reveal how models deliberate over temporal states, weigh possible interventions, and converge on action policies without human intervention. Evaluation emphasizes realized performance and reliability, commonly using cumulative reward, regret, policy value, Sharpe ratio, Sortino ratio, and control-specific key performance indicators.

**Advisory Decision Support.** The reasoning process is used to justify and rank candidate interventions, providing humans with transparent rationales and comparative analyses rather than raw predictions alone. Evaluation measures decision quality with users and adoption in practice, commonly using outcome improvement in user studies, choice consistency, and perceived usefulness of explanations.

### 2.4.4 Time Series Generation

This category concerns the direct creation or modification of temporal data. It includes simulation of synthetic series and scenario-driven generation where synthetic time series follow intended patterns. The goal is to learn a distribution $P_\theta$ that minimizes the divergence $\mathcal{D}$ from the true data distribution $P_{\text{data}}$, conditioned on specifications $C$ (prompts) and reasoning $\mathcal{Z}$:

$$\min_\theta \mathcal{D}(P_{\text{data}}(\mathbf{X} \mid C), P_\theta(\mathbf{X} \mid \mathcal{Z}, C, K)) \tag{7}$$

**Conditioned Synthesis.** Generative reasoning maps prompts or specifications into temporal dynamics, often requiring stepwise or branching inference to ensure the synthetic series follows intended trends or event patterns. Evaluation focuses on distributional fidelity, controllability, and diversity, commonly using maximum mean discrepancy and Kullback–Leibler divergence together with adherence to specified constraints.

These objective-level metrics primarily evaluate final task performance. They are necessary but not sufficient for evaluating TSR systems, since two methods can obtain similar output scores while differing in trace faithfulness, grounding, robustness under shift, cost, latency, and reproducibility. Section 7.1 therefore returns to complementary reasoning-level and topology-level evaluation criteria.

### 2.5 Attribute Tags

Beyond reasoning topology and primary objective, we record lightweight, non-exclusive attribute tags to capture additional properties of each work. These tags provide finer-grained descriptors and are grouped into

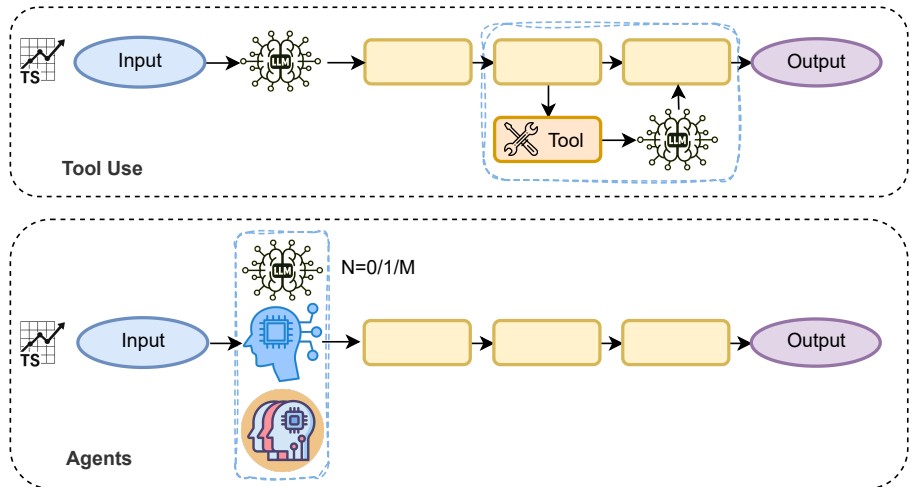

Figure 7: Execution actors: tool use, single-agent reasoning, and multi-agent reasoning.

four categories. Most tags are *binary*, meaning they are either present or absent in a given run. Only the agent tag and the LLM adaptation tag are *categorical*, requiring the assignment of exactly one value from a predefined set. Importantly, tags never override the reasoning topology: they describe auxiliary behaviors or properties observed in the execution trace, while the topology is always determined directly from the structure of that trace.

### 2.5.1 Control-Flow Operators

Control-flow operators characterize how the reasoning process is organized at the step-to-step level. The labeling rule is simple: we assign an operator tag only when its behavior is explicitly shown in the reasoning trace or method description, not when it is merely suggested by the task interface or stated as an intention. The control-flow operators we track are task decomposition, verification and critique, and ensemble selection, as illustrated in Figure 6.

**Task Decomposition.** Task decomposition is present when the method explicitly enumerates subproblems, subquestions, or subplans that structure subsequent execution. The operator is evidenced by visible subgoal statements or by a planner that emits discrete substeps used downstream. Task decomposition by itself does not determine the reasoning topology. When the subgoals are executed one after another in sequence, the resulting topology is linear chain reasoning. When multiple alternatives are explored, whether in parallel, independently, or with later feedback and recombination, the resulting topology is branch-structured reasoning. This outcome is independent of verification and critique, which may or may not be present as separate operators.

**Verification and Critique.** Verification and critique are present when there is an explicit step that evaluates candidate outputs or intermediate reasoning through judging, checking, critiquing, or self-refinement. Silent heuristics or implicit scoring internal to a single step without an externally visible judging action do not count as verification. The operator is evidenced by a visible critic, judge, or scoring step, which may be carried out by the same model, another model, or a human. If the evaluation is performed without inducing revisions, the reasoning topology remains unchanged and can be either direct reasoning, linear chain reasoning, or branch-structured reasoning, depending on the surrounding structure. When verification leads to regeneration, edits, or revisions to earlier content, the execution trace is branch-structured reasoning, since feedback creates additional paths or reconnects to previous ones. This includes multi-round self-refinement protocols, iterative critique-and-revise loops, and agent debates where arguments trigger revisions across

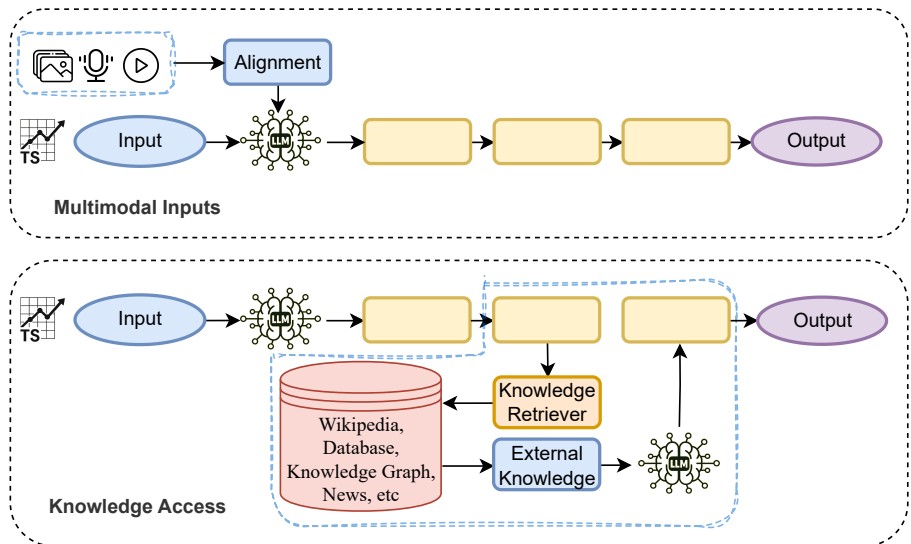

Figure 8: Information sources: multimodal inputs and external knowledge access.

steps. By contrast, single-round evaluation or one-shot debates that only select among existing candidates preserve the underlying topology without adding feedback or new branches.

**Ensemble Selection.**  Ensemble selection, often called self-ensemble or self-consistency in the LLM literature, is present when multiple candidate reasoning traces or predictions are explicitly compared and a final outcome is chosen by a rule such as voting, ranking, or averaging. When the candidates are produced by restarting the same procedure multiple times and then resolved only at the end, the trace remains linear chain reasoning. When multiple alternatives are maintained within a single run and later resolved by selection, ranking, or fusion, the trace is branch-structured reasoning.

### 2.5.2   Execution Actors

Execution actors specify the entities responsible for carrying out reasoning steps during execution, as illustrated in Figure 7. They indicate whether reasoning is performed solely by the model itself, delegated to external tools, or organized through autonomous agents that act at inference time.

**Tool Use.**  Tool use is present when the model invokes external resources such as search engines, solvers, or simulators during reasoning. The operator is evidenced by explicit calls to external systems whose outputs feed into subsequent reasoning steps. Tools are passive: they return information or computations but do not initiate new reasoning themselves.

**Agents.**  The agent tag captures cases where autonomous agents are present at inference time. An autonomous agent is a component that, given its current state, selects the next action or message in pursuit of a goal, often powered by an LLM and sometimes using tools or memory. This tag is categorical rather than binary: it records the number of agents, with possible values *0 = no agent*, *1 = a single agent*, and *M = multiple collaborating agents*. The overall reasoning topology is determined by how the agents interact. A one-round manager–worker handoff typically corresponds to linear chain reasoning, whereas scenarios involving multiple workers that propose alternatives and are later merged, or multi-round coordination and debate with feedback, are forms of branch-structured reasoning.

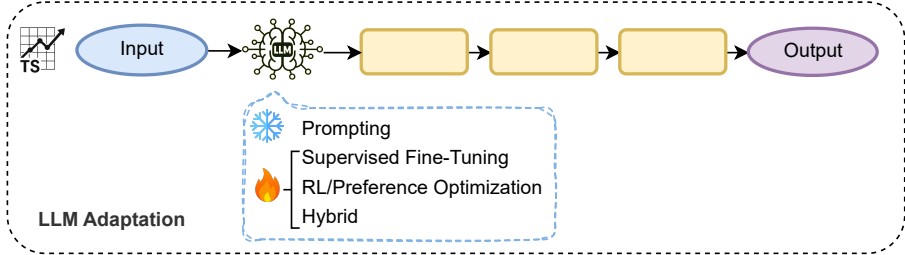

Figure 9: LLM adaptation regimes: prompting, supervised fine-tuning, reinforcement/preference optimization, and hybrid approaches.

### 2.5.3   Information Sources

Information sources capture inputs that extend beyond the raw time series itself, as illustrated in Figure 8. They cover both additional modalities, such as language or images, and external knowledge retrieved from databases, search engines, or domain resources.

**Multimodal Inputs.**   Multimodal inputs occur when time series are combined with other modalities such as natural language, images, audio, or structured reports. Such settings highlight scenarios where reasoning must integrate signals across different types of data rather than relying on temporal sequences alone.

**Knowledge Access.**   Knowledge access arises when the reasoning process incorporates external information through retrieval modules, search engines, structured databases, or domain-specific resources. It is evidenced by explicit calls to knowledge sources whose retrieved content conditions or supplements the model's reasoning.

### 2.5.4   LLM Adaptation Regimes

LLM adaptation regimes specify how large language models are trained or adapted to perform reasoning on time series tasks, as shown in Figure 9. This tag is categorical: exactly one regime is assigned to each method.

**Adaptation.**   The adaptation tag takes one of four regimes. *Prompting* relies on frozen models guided by instructions, few-shot examples, or chain-of-thought prompting without parameter updates. *Supervised fine-tuning* trains models on labeled temporal reasoning tasks such as instruction tuning, adapter methods, or distillation of reasoning traces. *Reinforcement or preference optimization* adapts models using feedback-based objectives, including reinforcement learning with human or AI feedback and preference optimization methods. *Hybrid approaches* combine supervised fine-tuning with reinforcement or preference optimization, for example by instruction-tuning a model before further aligning it with RLHF or direct preference optimization.

## 3   Direct Reasoning

Direct reasoning represents the most basic reasoning topology in the taxonomy. In this setting, a model directly maps time series inputs to outputs in a single step, without generating or exposing any intermediate reasoning trace. As such, direct reasoning can be viewed as the simplest baseline for time series reasoning: it provides efficiency and accessibility, but at the cost of limited interpretability and reduced robustness for complex tasks. Despite its simplicity, direct reasoning remains widely adopted in recent work, particularly for straightforward forecasting, anomaly detection, or descriptive question answering, and it often serves as a point of comparison for more structured reasoning topologies. The following discussion organizes direct reasoning methods according to four primary objectives, as illustrated in Figure 10.

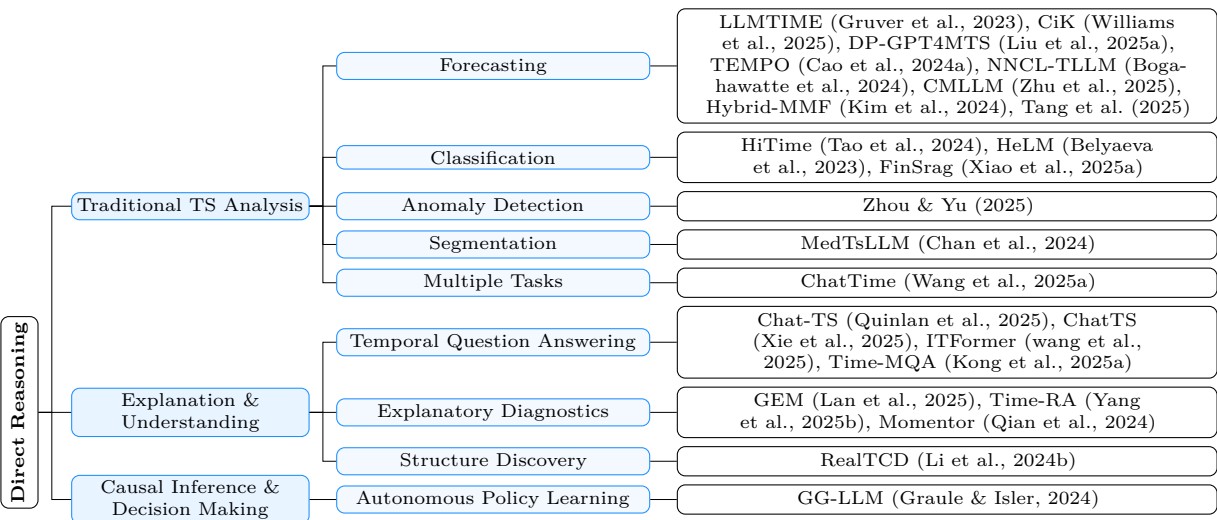

Figure 10: Taxonomy of direct reasoning approaches in time series reasoning

## 3.1 Traditional Time Series Analysis with Direct Reasoning

Traditional time series analysis under *direct reasoning* treats the model as a one-shot mapper from temporal inputs (optionally with side context) to outputs such as forecasts, class labels, segmentation masks, or anomaly intervals. The execution topology is a single forward generation or completion without an explicit, multi-step trace. Within this topology, recent work spans zero-shot prompting, parameter-efficient adaptation, multimodal fusion, and retrieval-augmented conditioning—while retaining a single-step inference interface.

**Forecasting.** LLMTIME (Gruver et al., 2023) reframes forecasting as next-token generation over textualized numbers, sampling multiple continuations to summarize point and probabilistic predictions while analyzing calibration, tokenization, and context-length effects. CiK (Williams et al., 2025) introduces a context-aided benchmark and evaluates a direct prompt that outputs structured probabilistic forecasts in one call, showing gains when textual context is informative but exposing occasional catastrophic failures. DP-GPT4MTS (Liu et al., 2025a) conditions a largely frozen GPT-2 backbone with dual prompts, consisting of an explicit instruction and statistics prompt together with a soft textual prompt derived from timestamped text, concatenated with patched time series embeddings to decode future values directly. TEMPO (Cao et al., 2024a) attaches component-specific prompts to decomposed trend, seasonality, and residual patches and fine-tunes GPT-2 with LoRA, predicting each component in a single forward pass and then additively combining them to form the forecast. NNCL-TLLM (Bogahawatte et al., 2024) learns time-series-compatible text prototypes and forms learned prompts through nearest-neighbor selection, while retaining a one-shot inference interface. These prompts are fed with patch embeddings into a partially tuned LLM that adjusts only positional embeddings and layer norms to generate the forecasts.

CMLLM (Zhu et al., 2025) converts wind turbine supervisory control and data acquisition (SCADA) signals into text, attaches a prior-knowledge prefix, and lets a frozen LLM generate forecast tokens that are projected back to numbers. Hybrid-MMF (Kim et al., 2024) jointly forecasts future numbers and narratives by aligning numeric and textual embeddings and decoding both modalities directly, and reports a negative result that highlights fusion challenges. Tang et al. (2025) studies simple prompt strategies by injecting human background knowledge or by reprogramming numeric histories into rise and fall prose, and reports consistent error reductions while noting difficulties on multi-period series.

**Classification.** HiTime (Tao et al., 2024) aligns time series and textual semantics so a tuned LLM generates class labels as text, improving both accuracy and F1 on UEA datasets in a single forward pass. HeLM (Belyaeva et al., 2023) maps spirogram waveforms and clinical variables into token space and computes label likelihoods such as asthma risk without intermediate planning, achieving strong AUROC and AUPRC on

UK Biobank traits. FinSrag (Xiao et al., 2025a) retrieves historical indicator segments, serializes them for prompting, and has a fine-tuned LLM directly predict stock movement as rise or fall in a single prompt call, improving both accuracy and MCC in financial forecasting.

**Anomaly Detection.** Zhou & Yu (2025) prompts LLMs and multimodal LLMs to return anomaly intervals from textualized sequences or plotted images in one step and finds that image inputs often outperform text, while subtle real-world anomalies remain challenging.

**Segmentation.** MedTsLLM (Chan et al., 2024) concatenates contextual text and signal patches (ECG, respiratory waveforms) into a frozen LLM and linearly projects output embeddings to produce segmentation masks, boundary points, or anomaly scores in one computation.

**Multiple Tasks.** ChatTime (Wang et al., 2025a) expands an LLM's tokenizer with discretized value symbols so a single model handles both time series forecasting and question answering, demonstrating transfer across tasks within a unified direct reasoning interface.

## 3.2 Explanation and Understanding with Direct Reasoning

This objective covers methods where the main product of reasoning is a natural-language answer, rationale, or causal interpretation derived from time series in a single inference step. The execution topology is direct: a model consumes temporal inputs (optionally fused with text or other modalities) and outputs explanatory text without iterative decomposition, branching, or explicit verification. Research here spans temporal QA, anomaly attribution, diagnostic reporting, and mechanism discovery.

**Temporal Question Answering.** Chat-TS (Quinlan et al., 2025) extends LLM vocabularies with discrete time series tokens and trains on multimodal instruction datasets to enable mixed time series text reasoning with direct answers and rationales while preserving general NLP ability. ChatTS (Xie et al., 2025) develops a multimodal LLM that integrates time series and text using synthetic QA generation and staged fine-tuning, enabling single-step explanation-oriented reasoning over trends, seasonality, anomalies, and causal queries. ITFormer (wang et al., 2025) freezes the backbone LLM and aligns temporal embeddings to the token space through a lightweight connector, enabling direct decoding of answers with strong efficiency and generalization demonstrated on new time series QA datasets. Time-MQA (Kong et al., 2025a) continually adapts LLMs on a large multi-domain QA corpus that unifies diverse time series tasks, enabling grounded and explanatory responses across forecasting, imputation, anomaly detection, classification, and open-ended reasoning.

**Explanatory Diagnostics.** GEM (Lan et al., 2025) aligns ECG waveforms, images, and text through frozen encoders and fine-tuned LLMs, introducing datasets and benchmarks that enable grounded diagnostic reports with clinician-style explanations. Time-RA (Yang et al., 2025b) introduces RATs40K, a large multimodal dataset for reasoning-centric anomaly detection where models generate observation–thought–action rationales in a single pass alongside detection, categorization, and explanatory reasoning refined through AI-feedback. Momentor (Qian et al., 2024) enhances video–LLMs with temporal token representations and event-sequence modeling, enabling segment-level localization and explanatory outputs in long untrimmed videos supported by a large-scale instruction dataset.

**Structure Discovery.** RealTCD (Li et al., 2024b) leverages an LLM to extract domain knowledge from textual system descriptions and propose candidate causal tuples that initialize a score-based causal discovery process. This meta-initialization step provides explanatory structure that guides the subsequent optimization of temporal causal graphs, while avoiding iterative reasoning during inference.

## 3.3 Causal Inference and Decision Making with Direct Reasoning

This objective concerns settings where the output is an action choice, policy signal, or quantified intervention effect derived from time series in a single inference step. Under direct reasoning, a model maps temporal

context (and optionally auxiliary descriptions or features) to a decision-relevant score or recommendation without intermediate steps or branching.

**Autonomous Policy Learning.** GG-LLM (Graule & Isler, 2024) presents a framework for human-aware robot task planning where a frozen LLM, prompted once with a narration of recent human activities, scores candidate interactions whose probabilities are geometrically grounded on a semantic map. A downstream planner uses these localized scores to guide robot coverage, reducing human disturbance by about 29% in simulated apartments. The work illustrates how a single language-model output can inform temporal planning while raising questions about robustness, probability calibration, and safety in embodied settings.

### 3.4 Attribute Tags with Direct Reasoning

#### 3.4.1 Control-Flow Operators with Direct Reasoning

**Task Decomposition.** Task decomposition is rarely adopted in direct reasoning and appears in only a small fraction of approaches. Examples include component-wise forecasting that predicts trend, seasonality, and residual components before summing them (Cao et al., 2024a), and structured output fields that separate observation, thought, and action for anomaly reasoning (Yang et al., 2025b).

**Verification and Critique.** Verification and critique are almost absent in direct reasoning and appear in only isolated cases. One example is the use of AI feedback to refine anomaly explanations during data construction, while inference remains single pass (Yang et al., 2025b).

**Ensemble Selection.** Ensemble selection is rarely employed in direct reasoning and appears in only a few approaches. One instance aggregates multiple forecast continuations by median or quantiles (Gruver et al., 2023), while another leverages outputs from a diverse model pool (GPT-4o, Gemini, DeepSeek-R1, Llama-3.3) ranked for reliability (Yang et al., 2025b).

#### 3.4.2 Execution Actors with Direct Reasoning

**Tool Use.** Tool use is almost absent in direct reasoning and is demonstrated in only a single work. An example is the retrieval of historical indicator segments injected into the prompt before a single LLM decision for financial forecasting (Xiao et al., 2025a).

**Agents.** Agents are not employed in direct reasoning, and all pipelines operate in a non-agentic manner.

#### 3.4.3 Information Sources with Direct Reasoning

**Multimodal Inputs.** The use of multimodal inputs is fairly common in direct reasoning and appears in a majority of approaches. Examples include combining time series with textual context for question answering and explanations (Quinlan et al., 2025; Xie et al., 2025; wang et al., 2025; Kong et al., 2025a), fusing health signals with text and tabular metadata for risk prediction (Belyaeva et al., 2023), and pairing signals with images such as ECG traces plus twelve-lead images (Lan et al., 2025). Other works render sequences as plots for visual anomaly prompts (Zhou & Yu, 2025), or use video as a temporal modality alongside text for fine-grained temporal understanding (Qian et al., 2024).

**Knowledge Access.** Knowledge access is almost absent in direct reasoning and is demonstrated in only a single work. One example is retrieval-augmented financial forecasting that conditions the LLM on retrieved historical patterns (Xiao et al., 2025a).

#### 3.4.4 LLM Adaptation Regimes with Direct Reasoning

**Adaptation.** Both prompt-only usage and supervised tuning are widely adopted in direct reasoning. Prompt-only approaches include forecasting, context-aided forecasting, video temporal reasoning, temporal causal initialization, and zero-shot analyses (Zhu et al., 2025; Williams et al., 2025; Gruver et al., 2023;

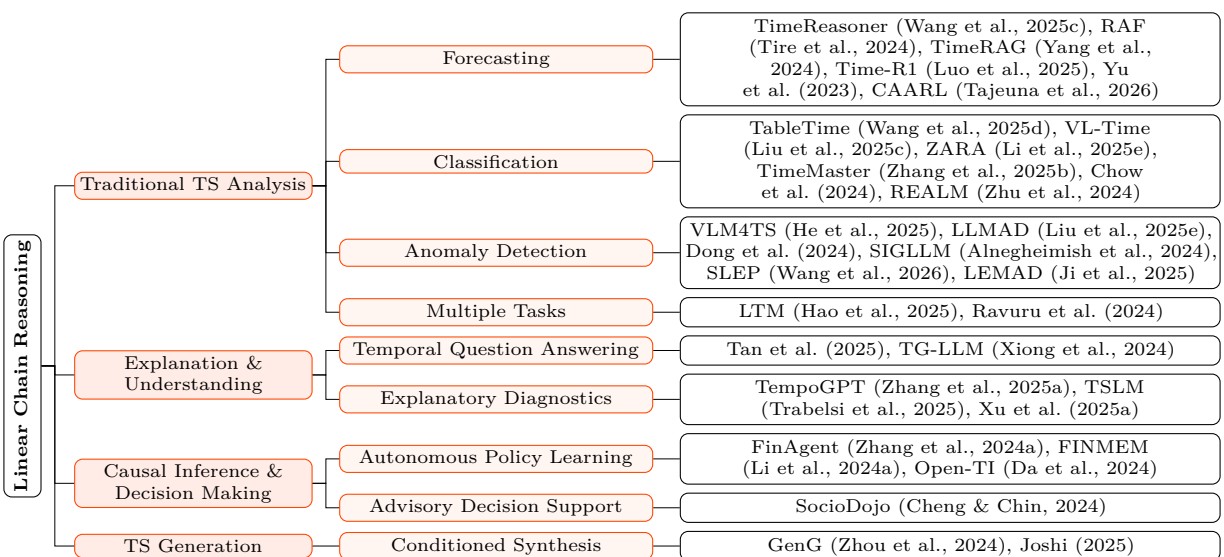

Figure 11: Taxonomy of linear chain reasoning approaches in time series reasoning

Qian et al., 2024; Li et al., 2024b; Tang et al., 2025). Instruction-tuned or adapter-based methods support multimodal reasoning, clinical interpretation, component-wise forecasting, joint numeric and text forecasting, health risk classification, and time-series question answering (Xie et al., 2025; Lan et al., 2025; Cao et al., 2024a; Kim et al., 2024; Belyaeva et al., 2023; Kong et al., 2025a). No reinforcement-only or hybrid regimes appear in this set.

## 3.5 Comparative Analysis

The primary advantage of direct reasoning is efficiency: with a single inference call, it offers the lowest latency and computational cost, making it well-suited for high-throughput applications such as real-time anomaly detection or streaming forecasting. However, this black-box nature is a significant limitation. Without visible intermediate steps, direct reasoning suffers from limited interpretability, since errors cannot be traced to a specific logical failure. It also exhibits reduced robustness, as the model cannot self-correct or backtrack once generation begins. Consequently, direct methods often serve as a lower-bound baseline, suitable for simpler tasks but frequently outperformed by more structured reasoning topologies on complex, multi-stage problems.

## 4 Linear Chain Reasoning

Linear chain reasoning denotes executions that proceed through a *single, ordered sequence of steps* with no in-trajectory branching. The model may explicitly decompose a task, invoke a tool or retrieval once, and optionally perform a *one-shot* verification pass, but it does not maintain multiple concurrent hypotheses or iterate critique–revise loops. This topology preserves much of the simplicity of direct reasoning while adding mild structure that can improve grounding and numerical stability, yet still avoids the latency and complexity of branch-structured systems. The following discussion organizes linear chain methods according to four primary objectives, as illustrated in Figure 11.

### 4.1 Traditional Time Series Analysis with Linear Chain Reasoning

Traditional time series analysis under *linear chain reasoning* implements scripted sequences such as *analyze → (retrieve) → predict* or *detect → verify → decide*, while retaining a single-path execution.

**Forecasting.** TimeReasoner (Wang et al., 2025c) treats time series forecasting as deliberate reasoning, using structured prompts so that LLMs analyze patterns before generating forecasts in a fixed linear sequence. RAF (Tire et al., 2024) introduces a retrieval-augmented framework for time series foundation models that builds dataset-specific databases, retrieves the most relevant temporal segments, and integrates them into the forecasting process, showing consistent improvements across diverse benchmarks. TimeRAG (Yang et al., 2024) proposes a retrieval-based approach that slices time series into representative segments, retrieves similar histories, and reprograms them into natural-language prompts for a frozen LLM, yielding forecasting gains on the M4 benchmark without modifying model weights. Time-R1 (Luo et al., 2025) aligns the scripted chain with supervised traces followed by preference and reinforcement learning optimization while keeping inference as a fixed, linear sequence. Yu et al. (2023) integrates historical prices with company profiles and news, prompting LLMs to first summarize and contextualize signals and then output forecasts and explanations in a fixed, linear sequence, with GPT-4 few-shot outperforming financial baselines. CAARL (Tajeuna et al., 2026) builds autoregressive sub-models and temporal dependency graphs, serializes them into natural-language scenarios, and uses in-context LLM reasoning to forecast co-evolving time series through a fixed sequential pipeline.

**Classification.** TableTime (Wang et al., 2025d) serializes time series into tabular prompts for training-free classification, using a fixed analyze-then-classify pass with optional self-consistency ensembles across runs and no in-trajectory branching. VL-Time (Liu et al., 2025c) renders time series as images and uses a plan-then-solve vision–language model to classify them, showing visual encoding overcomes tokenization limits of text-only LLMs. ZARA (Li et al., 2025e) performs zero-shot, classifier-free activity recognition by chaining feature-importance priors and multi-sensor retrieval into a serial LLM pipeline, with manager–worker handoffs yielding interpretable predictions. TimeMaster (Zhang et al., 2025b) trains a multimodal LLM with reinforcement learning to generate structured outputs over visualized time series data, combining reasoning, classification, and optional extension steps in a linear execution flow that boosts accuracy and context awareness. Chow et al. (2024) develops a multimodal LLM that aligns time series embeddings with a language model's token space and fine-tunes it on chain-of-thought–augmented tasks, improving recognition and reasoning performance while retaining a single-path execution style. REALM (Zhu et al., 2024) extracts disease entities from clinical notes, matches them to a knowledge graph, and fuses their embeddings with vital-sign time series in a linear RAG pipeline for clinical risk prediction.

**Anomaly Detection.** VLM4TS (He et al., 2025) introduces a two-stage anomaly detection framework that first screens candidate anomalies with a pretrained vision encoder and then verifies and refines them through a VLM, producing final decisions and explanations in a single pass rather than through iterative critique–revise loops. LLMAD (Liu et al., 2025e) proposes an LLM-based framework that retrieves similar series for in-context learning, injects domain knowledge, and applies a structured reasoning process to output anomaly points with explanations, achieving interpretable results without relying on repeated revision cycles. Dong et al. (2024) evaluates large language models as explainable anomaly detectors, showing that with carefully designed prompts or lightweight fine-tuning they can identify anomalies and provide explanations, but still face instability and hallucination issues, leading to a single candidate–verification flow rather than iterative regeneration. SIGLLM (Alnegheimish et al., 2024) explores zero-shot anomaly detection by converting numeric series to text and prompting LLMs either directly or via forecasting residuals, demonstrating competitive performance against classical baselines while following a straightforward candidate–reassessment path that retains linear execution. SLEP (Wang et al., 2026) presents an LLM-based agent for anomaly detection in power system time series that leverages structured prompts, series memory, and optional reflection to provide accurate judgments and concise explanations, keeping the verification step one-shot. LEMAD (Ji et al., 2025) proposes a hierarchical multi-agent framework where specialist agents collect metrics and parse logs and a manager agent fuses information to make global anomaly decisions, enabling interpretable root-cause explanations while maintaining a single verification stage.

**Multiple Tasks.** LTM (Hao et al., 2025) integrates a frozen LLM with a pre-trained time series model and knowledge-graph-driven prompts, using fusion and retrieval modules in a scripted linear pipeline to improve forecasting, imputation, and anomaly detection with minimal fine-tuning. Ravuru et al. (2024) presents a hierarchical multi-agent RAG framework where a master delegates time series tasks to specialized sub-agents

in a fixed sequence that retrieves dynamic prompts and improves forecasting, imputation, anomaly detection, and classification under distribution shifts, without in-run branching.

## 4.2 Explanation and Understanding with Linear Chain Reasoning

Explanation and understanding in the linear chain setting rely on a *single, ordered sequence* of analysis steps that culminate in an explanatory answer or narrative without maintaining concurrent alternatives or running critique–revise loops. Typical instances translate signals into structured intermediate forms (timelines, tables, or visualizations), optionally retrieve external knowledge, and then produce explanations, rationales, or summaries in one coherent trajectory.

**Temporal Question Answering.** Tan et al. (2025) infers natural-language event sequences that explain observed temporal segments by guiding stepwise analysis of changes and eliminating inconsistent options, producing a single answer in one path. TG-LLM (Xiong et al., 2024) translates narratives into aligned timelines and then executes deliberate reasoning over the structured representation to answer questions about order, duration, and simultaneity, while keeping verification for training rather than test-time branching.

**Explanatory Diagnostics.** TempoGPT (Zhang et al., 2025a) aligns temporal tokens and text in a shared space and trains the model to generate chain-of-thought rationales that culminate in conclusions such as trend analysis or fault diagnosis, executing one coherent trajectory per query. TSLM (Trabelsi et al., 2025) generates multiple local captions from a time series encoder–decoder and then consolidates them with a separate language model as a single end-of-chain summarization step, avoiding in-trajectory maintenance of alternatives. Xu et al. (2025a) orchestrates data acquisition, knowledge retrieval, analytic functions, and report writing through a serial controller for visual analytics, yielding grounded explanatory narratives and root-cause summaries without critique–revise loops at inference.

## 4.3 Causal Inference and Decision Making with Linear Chain Reasoning

Causal Inference and Decision Making in the linear chain setting executes a *single, ordered* observe→(retrieve)→decide pipeline, optionally with one-shot verification, to optimize a policy value over time. Typical instances ground actions with tools or memories and assess utility via returns, risk-adjusted metrics, or control rewards while avoiding in-run branching or debate.

**Autonomous Policy Learning.** FinAgent (Zhang et al., 2024a) sequences market intelligence, retrieval, immediate and high-level reflections, followed by a buy, sell, or hold action, achieving improved risk-adjusted returns from multimodal and tool-augmented inputs in a single path. FINMEM (Li et al., 2024a) sequences summarization, observation, retrieval, reflection, and decision within a layered memory system, extending reflection across days to adapt trading strategies while preserving a single-path execution. Open-TI (Da et al., 2024) integrates an LLM planner for configuring traffic simulations with a controller for signal actions, executing sequential thought-to-action steps to optimize throughput and travel time without branching.

**Advisory Decision Support.** SocioDojo (Cheng & Chin, 2024) coordinates analyst, assistant, and actuator roles to form hypotheses, retrieve evidence, and execute portfolio actions in a partially observable Markov decision process, using accept–reject as a one-shot verification step within a linear dialogue loop.

## 4.4 Time Series Generation with Linear Chain Reasoning

Time series generation in the linear chain setting follows a single, ordered script that first specifies targets or constraints and then executes a generator without maintaining concurrent alternatives or multi-round critique–revise loops. Typical instances use a language model to produce high-level descriptions or to guide tool configuration and data retrieval before a one-pass synthesis or consolidation step.

**Conditioned Synthesis.** GenG (Zhou et al., 2024) decomposes generation into a text-specification stage driven by a finetuned language model followed by conditional diffusion that synthesizes sequences under those

specifications, reporting improvements in fidelity, controllability, and downstream utility while preserving a fixed two-stage path. Joshi (2025) uses publicly available language models to derive trusted-domain queries and parameter settings that are human-checked once and then applied to train GAN and VAE generators on interest-rate series, with distributional comparisons and backtesting conducted after a single linear pipeline.

## 4.5 Attribute Tags with Linear Chain Reasoning

### 4.5.1 Control-Flow Operators with Linear Chain Reasoning

**Task Decomposition.** Task decomposition is a prevalent feature of linear chain reasoning, reported in the majority of works. Two-stage designs recur across the literature, including planning followed by solving in visualization-guided reasoning (Liu et al., 2025c) and in table-structured classification with ordered steps (Wang et al., 2025d), localization followed by verification for anomaly detection (He et al., 2025), translation into a temporal graph followed by reasoning (Xiong et al., 2024), and description of targets followed by time series generation (Zhou et al., 2024). Generation followed by reflection within a single trajectory appears in forecasting (Wang et al., 2025c), incorporates reassessment steps for anomaly detection decisions (Liu et al., 2025e), and is implemented through immediate or extended reflections in trading agents (Zhang et al., 2024a; Li et al., 2024a). Manager-to-worker handoffs without branching within the trajectory occur in operations pipelines and traffic-control toolchains (Ji et al., 2025; Da et al., 2024).

**Verification and critique.** Verification and critique are less common in linear chain reasoning and appear in a minority of works. Verification and critique mechanisms span both inference- and training-time safeguards. At inference, models employ self-reflection on prior outputs in trading and forecasting (Zhang et al., 2024a; Wang et al., 2025c; Li et al., 2024a); dedicated verifier modules refine candidate predictions in visual anomaly detection, and one-shot reassessment reduces false positives (He et al., 2025; Liu et al., 2025e). During training, judging and filtering—often via reward models—are used in temporal-graph reasoning and multimodal classification (Xiong et al., 2024; Zhang et al., 2025b). Complementary oversight includes human validation of query quality in generation frameworks (Joshi, 2025) and entity-level validation to filter hallucinations in clinical knowledge extraction (Zhu et al., 2024).

**Ensemble selection.** Ensemble selection is rarely used in linear chain reasoning and appears in only a few works. Self-consistency over repeated chains and aggregation of forecast samples both improve robustness, enhancing table-based classification (Wang et al., 2025d) and stabilizing zero-shot detectors (Alnegheimish et al., 2024). Generating multiple candidates followed by summarization consolidates caption candidates into a single description (Trabelsi et al., 2025).

### 4.5.2 Execution Actors with Linear Chain Reasoning

**Tool use.** Tool use is a frequent component of linear chain reasoning and features in numerous approaches. Knowledge retrieval and text tools span a range of applications, from knowledge graphs and retrieval-augmented generation for clinical prediction (Hao et al., 2025; Zhu et al., 2024), to web and news APIs for finance (Yu et al., 2023), long-term memory stores for trading (Zhang et al., 2024a), and vector databases that ground analytics pipelines (Xu et al., 2025a). Time series exemplar retrieval tools further extend these capabilities, leveraging DTW-based knowledge bases for forecasting (Yang et al., 2024), embedding retrieval for foundation forecasters (Tire et al., 2024), neighbor retrieval for classification (Wang et al., 2025d), and class-wise retrieval methods for activity recognition (Li et al., 2025e). Beyond retrieval, simulation and control tools are employed by traffic agents (Da et al., 2024). Finance and data connectors also underpin decision pipelines with retrieval and screening functions (Li et al., 2024a).

**Agents.** Most linear chain approaches operate without agents, reflecting this topology's dominance in the category (Liu et al., 2025c;e; Chow et al., 2024). Single-agent execution is occasionally adopted in domains such as trading and power system detection (Zhang et al., 2024a; Li et al., 2024a; Wang et al., 2026). Multi-agent coordination appears less frequently but is used for agentic RAG over time series tasks, traffic control, and zero-shot activity recognition (Ravuru et al., 2024; Da et al., 2024; Li et al., 2025e).

### 4.5.3 Information Sources with Linear Chain Reasoning

**Multimodal inputs.** The use of multimodal inputs is a frequent practice in linear chain reasoning and is present in many approaches. Incorporating text with time series signals is widespread, spanning trading that fuses prices with news (Zhang et al., 2024a), clinical prediction that blends notes with EHR series (Zhu et al., 2024), explainable stock forecasting that integrates company and macro news (Yu et al., 2023), and general time series reasoning that concatenates a dedicated series encoder with text for the LLM (Chow et al., 2024). Image or plot inputs are likewise fused with text, as in visualization-guided reasoning (Liu et al., 2025c), two-stage anomaly detection with a vision-language verifier (He et al., 2025), and structured reasoning over plotted series paired with textual prompts (Zhang et al., 2025b).

**Knowledge access.** Knowledge access recurs in linear chain reasoning and appears in a substantial share of approaches. Web and report retrieval actively conditions decisions in trading and portfolio studies (Zhang et al., 2024a; Yu et al., 2023; Cheng & Chin, 2024). Structured knowledge graphs steer entity-centric reasoning in both general and clinical settings (Hao et al., 2025; Zhu et al., 2024). Time series knowledge bases and prompt pools provide retrieved motifs and templates that guide downstream reasoning (Tire et al., 2024; Yang et al., 2024; Ravuru et al., 2024). Domain repositories and vector databases are used to ground both analytic workflows and anomaly-detection pipelines (Xu et al., 2025a; Liu et al., 2025e; Li et al., 2025e).

### 4.5.4 LLM Adaptation Regimes with Linear Chain Reasoning

**Adaptation.** Prompt-only adaptation is widely reported in linear chain reasoning and appears to be the dominant regime in the majority of approaches (Zhang et al., 2024a; Liu et al., 2025c;e; Da et al., 2024). Supervised tuning with instruction-based or adapter-style methods is also common, supporting anomaly detection with synthetic supervision (Dong et al., 2024), text-guided generation (Zhou et al., 2024), temporal graph reasoning (Xiong et al., 2024), and multimodal temporal language models (Zhang et al., 2025a). Hybrid pipelines that combine supervised and reinforcement components appear in a smaller subset of works, including agentic RAG over time series tasks, event inference from win probability series, slow-thinking forecasting with reinforced LLMs, and structured multimodal reasoning (Ravuru et al., 2024; Tan et al., 2025; Luo et al., 2025; Zhang et al., 2025b). No works in this set rely solely on reinforcement or preference optimization without supervision.

## 4.6 Comparative Analysis

Linear chain reasoning strikes a balance between the opacity of direct methods and the complexity of branching. Its key strength is modular interpretability: by exposing sequential steps (e.g., *Retrieve Data → Analyze Trend → Forecast*), it allows users to verify the logic and diagnose precisely where errors occur. However, this topology remains brittle due to error propagation. A single hallucination in an early step can cascade downstream, and the lack of branching prevents the model from recovering from such dead ends. Crucially, we distinguish this topology from direct reasoning, particularly in relation to Chain-of-Thought (CoT), based on token visibility. A method is classified as *linear chain* only when intermediate reasoning steps are explicitly materialized as discrete, observable output tokens or tool actions. In contrast, reasoning that remains internal to the model is categorized as *direct reasoning*.

## 5 Branch-Structured Reasoning

Branch-structured reasoning captures executions that *fork, revise, and fuse* intermediate hypotheses within a single run. A pipeline qualifies as branch-structured whenever it explores alternatives in parallel or sequentially, runs multi-round critique–revise loops, or aggregates across concurrent candidates; cross-branch fusion and debate-driven updates are canonical. Compared to linear chains, branching expands search and self-correction capacity but raises challenges in cost control, stability, and reproducibility. We organize branch-structured systems by primary objectives, as illustrated in Figure 12.

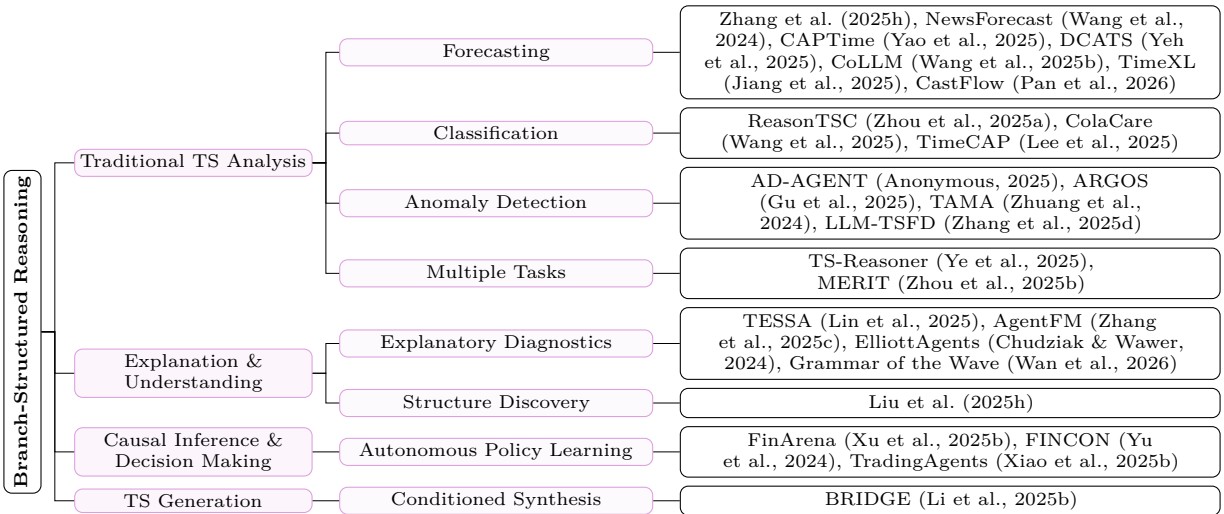

Figure 12: Taxonomy of branch-structured reasoning approaches in time series reasoning

## 5.1 Traditional Time Series Analysis with Branch-Structured Reasoning

**Forecasting.** Zhang et al. (2025h) frames news-driven forecasting as a competitive multi-agent process where parallel hypotheses are iteratively pruned and refined through self-reflection, with surviving agents' predictions aggregated into improved forecasts. NewsForecast (Wang et al., 2024) iterates reflection on errors against missed contextual factors and updates selection logic before regeneration, forming critique–revise loops that adjust earlier choices. CAPTime (Yao et al., 2025) routes among probabilistic experts with token-level fusion, enabling mixture-style decoding that reconciles concurrent generative paths and improves multimodal time series forecasting with a frozen LLM backbone. DCATS (Yeh et al., 2025) employs proposal–evaluate–refine cycles where an LLM agent iteratively selects and validates sub-datasets, improving data quality and forecasting accuracy across diverse backbone models. CoLLM (Wang et al., 2025b) routes predictions between small and large models using confidence scores, invoking stronger solvers only when needed and fusing uncertain outputs, achieving efficient and accurate remaining-life prediction. TimeXL (Jiang et al., 2025) couples a prototype-based encoder with prediction, reflection, and refinement agents that iteratively critique and revise forecasts, branching through feedback loops before fusing outputs into accurate, explainable time series predictions. CastFlow (Pan et al., 2026) learns a role-specialized agentic workflow for forecasting, combining planning, tool-based action, forecasting, and reflection with parallel exploration, ensemble aggregation, and reinforcement-based refinement.

**Classification.** ReasonTSC (Zhou et al., 2025a) conducts structured multi-turn reasoning that explicitly backtracks to explore alternatives before a fused decision, yielding internal branches that are reconciled. ColaCare (Wang et al., 2025) elicits divergent agent reviews with retrieved evidence and reconciles them through iterative debate and synthesis, generating fused clinical reports that improve predictions of mortality and readmission. TimeCAP (Lee et al., 2025) combines a contextualization–prediction branch driven by frozen LLMs with a multimodal encoder branch, fusing their outputs after independent reasoning to improve event prediction through cross-branch aggregation.

**Anomaly Detection.** AD-AGENT (Anonymous, 2025) decomposes anomaly detection into specialized agents coordinated by memory, iterating code and logic through generator–reviewer loops that revise upstream steps based on critiques and conditional retrieval. ARGOS (Gu et al., 2025) iteratively generates, repairs, and reviews detector rules with multiple agents, then fuses selected candidates with a base detector to yield more accurate and efficient anomaly detection. TAMA (Zhuang et al., 2024) converts series to images, analyzes across stages with a self-reflection pass that revises prior outputs, and applies sliding windows with pointwise voting for in-trajectory aggregation. LLM-TSFD (Zhang et al., 2025d) leverages human-in-the-loop

critique and knowledge-based corrections to iteratively refine pipelines, tools, and diagnostic explanations for industrial time series fault detection.

**Multiple Tasks.** TS-Reasoner (Ye et al., 2025) decomposes time series tasks into workflows of specialized operators and refines them with execution feedback, maintaining multiple candidate branches that are adaptively revised and fused for improved forecasting, anomaly detection, and causal discovery. MERIT (Zhou et al., 2025b) replaces hand-crafted augmentations with a multi-agent system that generates multiple candidate sequence views in parallel, verifies them, and selects the most suitable variants, yielding universal representations that enhance classification, forecasting, imputation, and anomaly detection across diverse datasets.

## 5.2 Explanation and Understanding with Branch-Structured Reasoning

Explanation and understanding in the branch-structured setting maintain multiple alternatives and allow critique–revise loops within a single run to reach an interpretable conclusion. Typical instances coordinate parallel roles or hypotheses and reconcile them through verification or fusion to produce diagnoses, symbolic rules, or explanatory annotations.

**Explanatory Diagnostics.** TESSA (Lin et al., 2025) coordinates a general annotator, a domain-specific annotator, and a reviewer that critiques and feeds back revisions, producing cross-domain explanatory annotations for time series through multi-agent branching and reconciliation. AgentFM (Zhang et al., 2025c) orchestrates system, data, and task agents over metrics, logs, and traces, aggregates divergent findings via a meta-agent, and delivers failure diagnoses and mitigation rationales by fusing parallel role-specific analyses. ElliottAgents (Chudziak & Wawer, 2024) explores stock time series through multiple candidate Elliott Wave patterns that are verified by deep reinforcement learning–based backtesting and synthesized into explanatory wave-structured reports, with agents branching across detection, validation, and reporting before convergence. Grammar of the Wave (Wan et al., 2026) introduces neuro-symbolic VLM agents for multivariate event detection, decomposing natural-language event descriptions into symbolic event logic trees and iteratively grounding them in signal visualizations with tool-assisted refinement.

**Structure Discovery.** Liu et al. (2025h) iteratively generates, evaluates, and refines symbolic structures for time series dynamics through propose–verify–refine loops, producing interpretable equations and causal rules validated by quantitative metrics and rubric-based checks.

## 5.3 Causal Inference and Decision Making with Branch-Structured Reasoning

Causal Inference and Decision Making in the branch-structured setting maintain multiple alternatives, run critique–revise loops, and fuse candidate plans to select actions over time. Typical pipelines coordinate specialized roles, retrieve external knowledge, and reconcile disagreements via debate or confidence-aware fusion, with evaluation by policy metrics such as cumulative/annual return, Sharpe ratio, or drawdown.

**Autonomous Policy Learning.** FinArena (Xu et al., 2025b) coordinates agents for time series, news, and statements, aggregates their outputs with user risk preferences, and produces personalized trading actions, combining parallel analysis, adaptive retrieval, and iterative reasoning to improve financial decision making. FINCON (Yu et al., 2024) coordinates analyst and manager agents with dual-level risk control that critiques trajectories and updates beliefs, producing portfolio policies optimized for returns, Sharpe ratio, and drawdown in partially observable financial markets. TradingAgents (Xiao et al., 2025b) conducts multi-round debates between bullish and bearish researchers with a facilitator selection and a risk team's adjustments, integrating tool-augmented retrieval and producing day-by-day trading decisions evaluated by backtest returns and risk.

### 5.4 Time Series Generation with Branch-Structured Reasoning

Time series generation in the branch-structured setting maintains multiple alternatives, runs critique–revise loops, and fuses candidates to produce controllable synthetic sequences. Typical instances coordinate parallel roles and reconcile textual or symbolic specifications before conditioning a generator in a single execution.

**Conditioned Synthesis.** BRIDGE (Li et al., 2025b) maintains parallel agent teams that iteratively propose, critique, and refine textual descriptions for target series, retrieve external evidence, and reconcile alternatives via consensus to produce conditioning inputs. The refined descriptions are then fused with learned temporal prototypes and a frozen text encoder to condition a diffusion generator, yielding controllable synthesis evaluated by fidelity and adherence metrics across diverse domains. This branching procedure improves semantic controllability over single-path baselines, albeit at the cost of additional compute for iterative refinement.

### 5.5 Attribute Tags with Branch-Structured Reasoning

#### 5.5.1 Control-Flow Operators with Branch-Structured Reasoning

**Task Decomposition.** Task decomposition is almost universal in branch-structured reasoning, appearing in most systems. Role-specialized stages, where distinct agents or modules own complementary subgoals, are standard (Xu et al., 2025b; Yu et al., 2024; Lin et al., 2025; Zhang et al., 2025c). Propose–repair–review loops that enumerate candidates, then iteratively fix and reassess, are also frequent (Gu et al., 2025). Hierarchical orchestration with a manager that integrates or routes among workers appears in several systems (Xu et al., 2025b; Zhang et al., 2025c). Multi-round discussions where branches debate and a coordinator reconciles outcomes further illustrate decomposition with feedback (Xiao et al., 2025b).

**Verification and Critique.** Verification and critique are prominent elements of branch-structured reasoning, reported in most literature. Self-reflection that checks predictions or intermediate text and then triggers revisions is explicit in closed-loop designs (Jiang et al., 2025). Code-level review and repair, where candidate detection rules are debugged and refined through explicit repair and review agents, occurs in rule-generation pipelines (Gu et al., 2025). Reviewer modules or human-in-the-loop checks that send feedback upstream are used to refine annotations and decisions (Lin et al., 2025; Xu et al., 2025b). Debate-style critique, where opposing branches argue and a facilitator selects or adjusts the plan, is another recurring pattern (Xiao et al., 2025b).

**Ensemble Selection.** Within branch-structured reasoning, ensemble selection is relatively uncommon, described in a smaller subset of the literature. Top-$k$ selection with later fusion of alternatives is used in rule-based anomaly detection (Gu et al., 2025). Late fusion that combines encoder outputs with LLM predictions within the same run is used in multimodal forecasting pipelines (Jiang et al., 2025; Lee et al., 2025).

#### 5.5.2 Execution Actors with Branch-Structured Reasoning

**Tool Use.** Tool use emerges as a prominent element of branch-structured reasoning and is present in many published approaches. Systems call external retrieval over the web and market providers (Xu et al., 2025b; Xiao et al., 2025b), optimization or analytics components such as portfolio solvers and risk calculators (Yu et al., 2024), and code execution or indicator calculators inside the loop (Xiao et al., 2025b).

**Agents.** Multi-agent execution with coordinated specialists is the dominant pattern in branch-structured reasoning, appearing in the majority of works (Xu et al., 2025b; Yu et al., 2024; Zhang et al., 2025c; Lin et al., 2025). Single-agent branches with iterative self-refinement are occasionally reported (Yeh et al., 2025), while some approaches operate without explicit agent components (Zhuang et al., 2024).

### 5.5.3 Information Sources with Branch-Structured Reasoning

**Multimodal Inputs.** The use of multimodal inputs is a common practice in branch-structured reasoning, reported in many works. Time series combined with text is common in forecasting and analysis frameworks (Jiang et al., 2025; Lee et al., 2025; Xu et al., 2025b; Lin et al., 2025). Some works also incorporate audio transcripts alongside market time series and text (Yu et al., 2024). Visual inputs such as time series plots, when treated as an image modality for MLLMs, are used in symbolic reasoning pipelines (Liu et al., 2025h).

**Knowledge Access.** Knowledge access is fairly common in branch-structured reasoning and appears in a majority of reported approaches. Adaptive web search and provider APIs supply external evidence that conditions downstream reasoning (Xu et al., 2025b; Xiao et al., 2025b). Other pipelines retrieve domain documents or stored memories to ground decisions (Yu et al., 2024; Zhou et al., 2025b), and retrieve in-context examples for augmentation (Lee et al., 2025).

### 5.5.4 LLM Adaptation Regimes with Branch-Structured Reasoning

**Adaptation.** Prompt-only usage with frozen backbones remains the dominant regime in this bucket (Gu et al., 2025; Lin et al., 2025; Jiang et al., 2025; Xu et al., 2025b; Lee et al., 2025; Zhang et al., 2025c; Xiao et al., 2025b; Wan et al., 2026). Supervised fine-tuning also appears in a few task-specific systems (Zhang et al., 2025h; Wang et al., 2025b; 2024), while hybrid supervised and reinforcement-style optimization appears in workflow-trained forecasting systems such as CastFlow (Pan et al., 2026).

## 5.6 Comparative Analysis

Branch-structured reasoning offers the highest theoretical ceiling for performance, particularly in ambiguous or high-stakes scenarios. By exploring multiple hypotheses in parallel and aggregating results, it drastically reduces the variance associated with single-path generation and enables self-correction by pruning incorrect branches before they corrupt the final answer. This makes it the most robust choice for complex diagnostics or volatile market trading, where a single perspective is insufficient. The trade-off is high computational cost and latency. Maintaining multiple agents or trajectory trees can scale linearly or exponentially with reasoning breadth and depth. Furthermore, the complexity of orchestrating multiple context windows increases the risk of coordination failures. Therefore, branch-structured designs are currently justified primarily when the cost of an error is high and the inference time budget permits iterative deliberation.

## 6 Current Landscape and Resources

This section surveys key resources for time series reasoning, focusing on datasets and benchmarks that vary in how directly they test reasoning, on surveys and position papers that synthesize progress and outline research agendas, and on recent studies that critically examine model performance and generalization. Together, these components provide a comprehensive view of both the resources that enable research and the evidence that shapes current understanding. This organization is summarized in Figure 13.

## 6.1 Datasets and Benchmarks

We group datasets and benchmarks by how directly they test time series reasoning. First, reasoning-first benchmarks define tasks and splits that explicitly require skills such as feature understanding, compositional generalization, temporal question answering, intervention reasoning, or agentic planning. In contrast, reasoning-ready benchmarks were not built primarily for reasoning but naturally support it through aligned side information, chronological or event-aligned protocols, or other structures that can be prompted into reasoning tasks. Finally, general-purpose time series benchmarks are standard collections for forecasting, detection, imputation, and related tasks that do not target reasoning by default, yet serve as solid references and can be adapted with minimal modification.

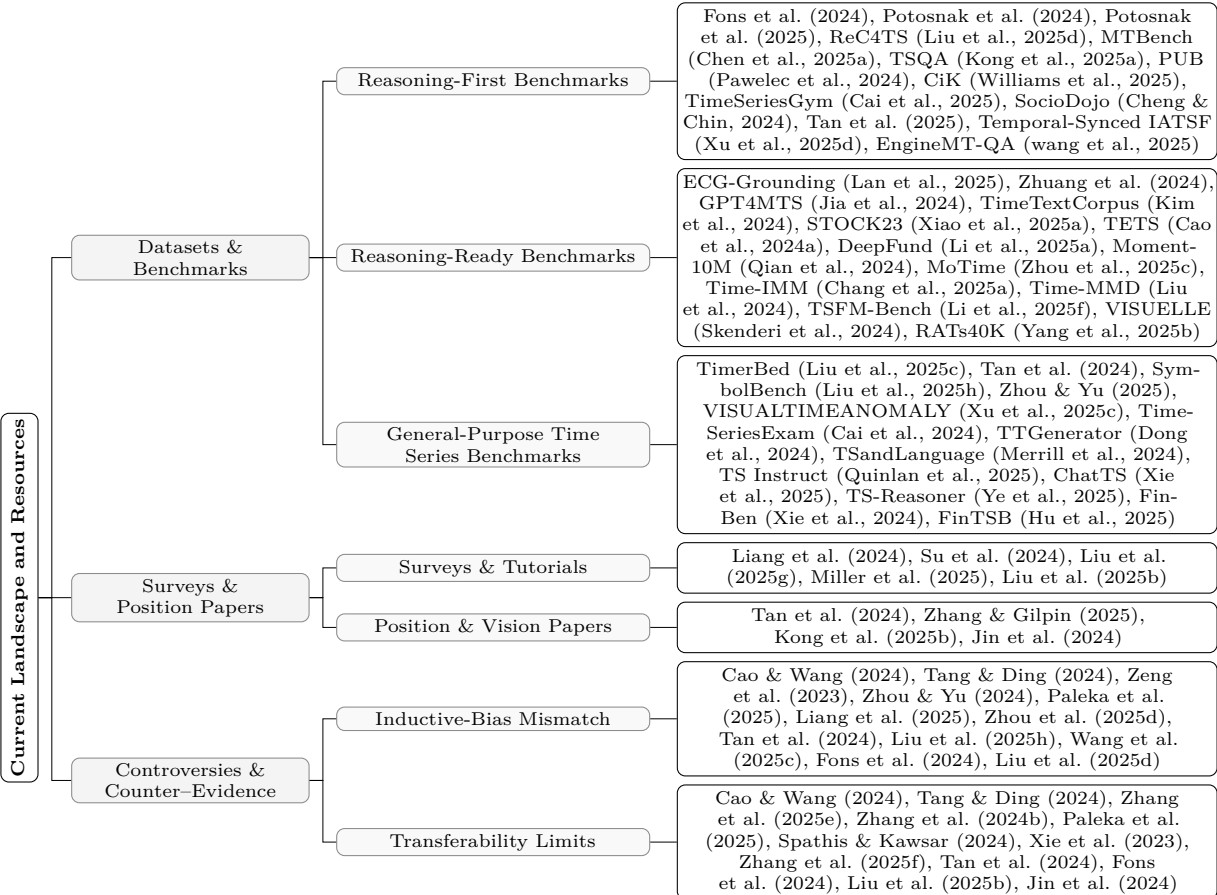

Figure 13: Taxonomy of current landscape and resources in time series reasoning

### 6.1.1   Reasoning-First Benchmarks.

Fons et al. (2024); Potosnak et al. (2024; 2025) introduce synthetic evaluations that test feature understanding and compositional generalization through controlled templates and held-out compositions, while ReC4TS (Liu et al., 2025d) develops a complementary evaluation targeting reasoning strategies and test-time sampling in time series forecasting. MTBench (Chen et al., 2025a) and TSQA (Kong et al., 2025a) frame temporal question-answering over time series: MTBench emphasizes cross-modal QA that links textual reports with series to test semantic trend understanding, indicator prediction, and correlation-based questions, while TSQA frames a broad set of time-series tasks as natural-language QA with prompts and rationale generation to elicit stepwise temporal reasoning. PUB (Pawelec et al., 2024) introduces a synthetic plot-understanding evaluation where generated charts, including time series plots with anomalies and degradations, are paired with JSON-structured questions to test visual and temporal reasoning. CiK (Williams et al., 2025) introduces context-dependent forecasting tasks that pair time series histories with textual cues and evaluates models by comparing performance with context present versus withheld, using a context-weighted CRPS-style scoring rule and prompting baselines for comparison. TimeSeriesGym (Cai et al., 2025) and SocioDojo (Cheng & Chin, 2024) construct episodic, reproducible environments for evaluating agentic decision making over time series, covering planning, tool use, iterative refinement, and seeded runs for systematic comparison. Tan et al. (2025), Temporal-Synced IATSF (Xu et al., 2025d), and EngineMT-QA (wang et al., 2025) align time-indexed series with textual descriptions of events or interventions and use release-aligned or event-aligned protocols, so models must infer timing, effect windows, and procedure-aware dynamics.

### 6.1.2   Reasoning-Ready Benchmarks.

ECG-Grounding (Lan et al., 2025) and Zhuang et al. (2024) pair clinical signals with contextual records and use case- or patient-level evaluation to support diagnostic and temporal reasoning under clinician-like constraints. GPT4MTS (Jia et al., 2024), TimeTextCorpus (Kim et al., 2024), STOCK23 (Xiao et al., 2025a), and TETS (Cao et al., 2024a) couple time series with news, descriptions, or structured text for context-aware forecasting and use chronology-preserving evaluations with ablations that toggle auxiliary context to isolate reasoning over external information, while DeepFund (Li et al., 2025a) provides a related live evaluation of model workflows on streaming market data but does not primarily target text-ablation protocols. Moment-10M (Qian et al., 2024) provides temporal localization with aligned segment boundaries that enable fine-grained reasoning over time, while MoTime (Zhou et al., 2025c) supplies multimodal, description-aligned time series with standardized splits and protocols for alignment-based forecasting and cold-start evaluation. Time-IMM (Chang et al., 2025a), Time-MMD (Liu et al., 2024), and TSFM-Bench (Li et al., 2025f) assemble large, multi-domain suites with unified loaders, canonical splits, and baseline code to facilitate controlled, reproducible reasoning studies at scale. VISUELLE (Skenderi et al., 2024) targets cold-start demand forecasting by combining product metadata and images with exogenous time series signals, and uses release-based splits to enforce evaluation under information asymmetry. RATs40K (Yang et al., 2025b) provides anomaly-focused multimodal resources built with critique-and-revise label construction and protocols that evaluate detection, localization, categorization, and explanatory reasoning across disturbance regimes and data sources.

### 6.1.3   General-Purpose Time Series Benchmarks.

TimerBed (Liu et al., 2025c) and Tan et al. (2024) provide stratified benchmarks and cross-dataset evaluations that use standardized chronological partitions, serving as reference points for comparing forecasting and LLM-based methods across domains. SymbolBench (Liu et al., 2025h), Zhou & Yu (2025), VISUALTIMEANOMALY (Xu et al., 2025c), and TimeSeriesExam (Cai et al., 2024) supply controllable, reproducible suites with fixed seeds, perturbation controls, and procedurally generated scenario catalogs that enable exact reproducibility for forecasting, anomaly, symbolic, and reasoning evaluations. TTGenerator (Dong et al., 2024) curates anomaly and change-point resources with event and tolerance-window protocols that emphasize temporal localization, while TSandLanguage (Merrill et al., 2024) provides a broader reasoning-focused suite including etiological reasoning, question answering, and context-aided forecasting using large-scale multiple-choice evaluation rather than change-point localization. TS Instruct (Quinlan et al., 2025) and ChatTS (Xie et al., 2025) provide multimodal classification and segmentation resources for sensor and biomedical streams and use subject-wise or group-wise evaluation splits when applicable to assess generalization across entities. TS-Reasoner (Ye et al., 2025) proposes an LLM-driven agentic framework that combines imputation and forecasting under prescribed masking patterns and rolling chronological splits to evaluate reconstruction and next-step prediction under missingness, using execution-feedback self-refinement and specialized operators to improve numeric fidelity. FinBen (Xie et al., 2024) aggregates finance time series with leakage-aware backtesting protocols oriented toward realistic temporal deployment. FinTSB (Hu et al., 2025) assembles financial forecasting testbeds with fixed temporal ranges, leakage-aware splits, and realistic backtesting suitable for standardized comparisons.

## 6.2   Surveys and Position Papers

The literature in this area includes both surveys and tutorials as well as position and vision papers. Surveys and tutorials synthesize progress by mapping methods, datasets, and open challenges in time series reasoning, offering structured overviews that guide researchers and practitioners. Position and vision papers, on the other hand, put forward arguments for or against specific approaches, outline perspectives on emerging opportunities, and set research agendas for the field.

### 6.2.1   Surveys and Tutorials.

Liang et al. (2024); Su et al. (2024) map the landscape of foundation models and LLM applications for time series, organizing architectures, pretraining and adaptation regimes, datasets, and task coverage. They

identify gaps in robustness, multimodality, and evaluation practice, motivating evaluation that prioritizes reasoning and robustness rather than incremental accuracy gains. Liu et al. (2025g) surveys synthetic data for time series and organizes the literature by generation methods and lifecycle use in pretraining, finetuning, and evaluation. It explains why particular design choices make synthetic corpora useful for probing feature understanding, compositional behavior, and temporal consistency, and it highlights limitations and future directions. Miller et al. (2025); Liu et al. (2025b) provide tutorials that unify common pipelines across forecasting, detection, classification, and analytics. They curate representative methods and reading paths and discuss pitfalls such as data leakage and misaligned splits that can obscure or mimic reasoning signals.

### 6.2.2 Position and Vision Papers.

Tan et al. (2024); Zhang & Gilpin (2025) question default uses of large language models for time series by presenting extensive replications and simple context-based baselines. They argue that progress should target settings where structured reasoning and external context matter and call for protocols that make those requirements explicit. Kong et al. (2025b); Jin et al. (2024) propose research agendas that emphasize compositional generalization, causal grounding, multimodal integration, and transparent evaluation. They advocate benchmarks and tools that reveal when a model is reasoning, when it is copying context, and when it fails under distribution shift.

## 6.3 Controversies and Counter–Evidence

A complementary line of research reports cases where Transformer and large language model methods for time series need careful interpretation. Two recurring themes appear across replication studies, ablation analyses, and position papers. The goal is to improve evaluation practices and clarify the limits of current approaches without changing the overall focus of the survey.

### 6.3.1 Inductive-Bias Mismatch.

A consistent observation is that generic Transformer and large language model backbones often lack the inductive structure that many time series tasks require, including clear seasonality, multiple repeating patterns, strong local correlations, and shifts between regimes Cao & Wang (2024); Liang et al. (2025); Zhou et al. (2025d); Liu et al. (2025h). Across forecasting and related tasks, several studies report results similar to tuned classical or specialized time series methods, and in many cases those baselines perform better, especially when horizon-wise evaluation is used with careful control of splits and data leakage Tan et al. (2024); Zeng et al. (2023); Zhou & Yu (2024); Wang et al. (2025c). The gap is most visible for long-horizon prediction and for series with strong frequency patterns Zeng et al. (2023); Wang et al. (2025c); Liu et al. (2025d); Zhou et al. (2025d). These results suggest that some reported improvements linked to reasoning or longer context are actually affected by missing time series priors when baselines and reporting methods are not carefully designed Tan et al. (2024); Tang & Ding (2024); Fons et al. (2024); Paleka et al. (2025). In practice, horizon-based reporting and strong baseline selection are often needed to interpret gains reliably compared to established time series methods Tan et al. (2024); Tang & Ding (2024); Fons et al. (2024); Paleka et al. (2025). When gains are observed, they often appear alongside components that include decomposition, frequency awareness, state-space modeling, or convolutional structures Cao & Wang (2024).

### 6.3.2 Transferability Limits.

A second line of evidence links mixed or negative results to a mismatch between pretraining priors and downstream time series dynamics, as well as to weak transfer across datasets and domains Tan et al. (2024); Tang & Ding (2024); Fons et al. (2024); Paleka et al. (2025). Pretraining on text and code does not by itself provide models with strong numeric accuracy, periodic structure, or stability over time Zhang et al. (2025e); Spathis & Kawsar (2024); Zhang et al. (2025f); Liu et al. (2025b). Reported gains can depend on dataset characteristics, prompt design, or evaluation choices that do not hold under domain shift or cross-benchmark testing Fons et al. (2024); Liu et al. (2025b); Jin et al. (2024); Xie et al. (2023). Replications document rank reversals across datasets, sensitivity to prompt wording, shot order, and random seed, and performance drops under shift or cold start settings Cao & Wang (2024); Fons et al. (2024); Zhang et al. (2024b); Liu et al.

| Topology | Inspectable evidence | Robustness pattern | Cost/latency | Reproducibility requirements |
|---|---|---|---|---|
| Direct | Final output only; failures are difficult to localize. | Often stable and efficient for simple tasks, but lacks visible self-correction under ambiguity or shift. | Low. | Fixed model, prompt, decoding settings, data splits, and preprocessing. |
| Linear chain | Ordered intermediate steps, retrieved evidence, and tool calls. | Supports stepwise diagnosis and grounding, but early hallucinations or retrieval errors can propagate downstream. | Medium. | Step templates, deterministic settings where possible, tool versions, seeds, and intermediate traces. |
| Branch-structured | Alternative hypotheses, critiques, revisions, and aggregation decisions. | Can reduce single-path variance and support self-correction, but is sensitive to search depth, pruning, aggregation, and tool drift. | High. | Branch logs, branch budgets, pruning rules, stopping criteria, seeds, model/tool versions, and aggregation policy. |

Table 1: Evaluation-oriented comparison of reasoning topologies. The table consolidates the tradeoffs discussed in Sections 3–5 and highlights what must be measured or reported for robust comparison.

(2025b). These findings indicate that adaptation, such as small parameter-efficient finetuning or numerically aware output heads, and drift-aware protocols, are prerequisites for assessing generalization Tan et al. (2024); Zhang et al. (2025e); Xie et al. (2023); Zhang et al. (2025f). Together they motivate reporting that includes leave-one-dataset-out in addition to averages, versioned splits with fixed seeds, explicit leakage checks, and calibration or coverage evaluations when claims involve transfer, cross-domain utility, or robustness Paleka et al. (2025).

## 7 Open Problems and Outlook

We highlight six themes that recur across direct, chain, and branch topologies, reflecting open problems that are repeatedly emphasized in author-stated outlooks and pointing to areas where further progress is needed.

### 7.1 Evaluation and Benchmarking.

Many works call for standardized, stress-tested evaluation rather than small curated sets, which often mean narrow domains with few series or short horizons, handpicked or templated prompts, pre-segmented windows that hide boundary effects, single-source cohorts, and limited resampling or shift tests (Liang et al., 2024; Su et al., 2024). To move beyond these limitations, stronger practice should include versioned splits with release-aligned protocols, fixed seeds, and reporting templates for reproducibility, as well as human studies where domain judgment matters (Tan et al., 2024; Fons et al., 2024; Lan et al., 2025).

Evaluation should separate three layers. First, output-level metrics measure final task performance, such as forecasting error and calibration, classification or anomaly F1, event localization, policy value, regret, and distributional fidelity, as summarized by objective in Section 2.4. Second, reasoning-level metrics test whether intermediate traces are faithful to the temporal evidence. This includes sufficiency and necessity of cited evidence, agreement between explanations and underlying data, temporal localization accuracy, and consistency between rationales and numeric outputs (Dong et al., 2024; Yao et al., 2025; Yeh et al., 2025; Wang et al., 2024; Li et al., 2025e). Post-hoc grading and LLM-as-judge approaches, while useful, are not sufficient unless paired with groundable signals, audit trails, or counterevidence checks (Liu et al., 2025h; Quinlan et al., 2025). Third, topology-level metrics evaluate how execution structure behaves under realistic stress, including horizon shift, domain shift, missingness, perturbations, prompt or seed sensitivity, latency, tool spend, and stability under different branch budgets or stopping rules.

The representative comparisons behind Table 1 recur across several application areas. In anomaly detection, direct visual or textual prompting can localize obvious intervals efficiently but often leaves failures hard to diagnose (Zhou & Yu, 2025); linear pipelines add candidate screening, retrieval, or one-shot verification to make false positives more inspectable (He et al., 2025; Liu et al., 2025e; Dong et al., 2024); and branch systems use generator–reviewer or repair loops to revise detector logic, improving auditability while making results depend on branch budgets, tools, and stopping rules (Anonymous, 2025; Gu et al., 2025). In context-aware forecasting, direct context prompts and textualized-number models test whether side information helps in one call (Williams et al., 2025; Gruver et al., 2023), retrieval or decomposition chains expose the selected evidence before prediction (Wang et al., 2025c; Tire et al., 2024; Yang et al., 2024), and branch or reflection systems explore competing event interpretations before aggregating predictions (Zhang et al., 2025h; Wang et al., 2024; Pan et al., 2026). In decision support, one-shot policy scoring is low-latency but offers little protection against spurious context, whereas memory/reflection chains and multi-agent debate systems surface assumptions, risk checks, and disagreements before actions (Zhang et al., 2024a; Li et al., 2024a; Xu et al., 2025b; Yu et al., 2024; Xiao et al., 2025b). Across these examples, the taxonomy is useful less as a performance ranking than as a diagnostic lens: it identifies where evidence enters, where errors can be checked, and which evaluation controls are needed. Current cross-paper evidence remains weak when methods are tested on different datasets, shifts, prompt budgets, or tool stacks, so topology-level robustness claims should be treated as hypotheses to be stress-tested rather than settled conclusions.

Branch-structured systems require especially careful reporting because the final output can depend on a search process rather than a single trajectory. Evaluations should specify the branch budget, pruning rule, stopping criterion, aggregation policy, random seeds, model versions, tool or API versions, and whether full branch traces are released. Otherwise, apparent gains may reflect uncontrolled search depth, favorable sampling, or tool-version drift rather than a robust reasoning advantage. More broadly, current evidence does not show that greater topological complexity automatically improves robustness: topology changes what can be inspected, revised, and aggregated, but correctness still depends on task-specific metrics, stress tests, and reproducible execution logs.

Looking ahead, evaluation suites should connect intermediate reasoning to user value, through measures such as decision impact, policy loss, constraint violations, intervention safety, latency, and tool spend, rather than relying only on proxy scores (Hu et al., 2025; Xie et al., 2024). While forecasting and causal tasks already benefit from multiple datasets and benchmarks, related areas such as time series editing—repair, counterfactual editing, and constrained rewriting—remain underrepresented. More explicit task definitions, faithful scoring protocols, and stress-tested datasets will be key to advancing this frontier (Liu et al., 2025c; Zhou et al., 2025a; Fons et al., 2024).

## 7.2   Multimodal Fusion and Cross-Modal Alignment.

Improving the alignment between time series, text, and images or videos remains a central challenge (Quinlan et al., 2025; Williams et al., 2025; Liang et al., 2024). Alignment spans three levels: instance pairing across modalities for the same example, temporal correspondence that ties tokens or pixels to the correct time indices, and semantic grounding that links claims to the measured signal (Liang et al., 2024). Fine-grained temporal localization requires identifying the exact time point or interval in the series that a word, phrase, or visual element refers to, and verifying that the predicted span matches the ground truth at the correct scale (Yeh et al., 2025; Xie et al., 2024). Further progress calls for stronger connectors and cross-modal objectives that explicitly link time indices to tokens or pixels. Promising directions include contrastive training with hard near misses (Liu et al., 2024), segment-level alignment losses (Yeh et al., 2025), pointer-style time-span decoders (Qian et al., 2024), and learned similarity beyond DTW (Yang et al., 2024).

Modality imbalance is common because the text channel often provides more tokens and denser labels, while the time series offers weaker supervision and more noise. This imbalance can cause models to overfit to text and overlook subtle temporal changes (Zhou et al., 2025c; Williams et al., 2025; Skenderi et al., 2024). Mitigation strategies include balanced sampling and loss reweighting across modalities, modality dropout and learned gating, per-modality normalization, and distillation that transfers signal from the stronger to the weaker modality (Zhou et al., 2025c; Williams et al., 2025; Belyaeva et al., 2023; Liu et al., 2025a).

Temporal synchronization across modalities is another frequent failure mode. Differences in sampling rates, logging delays, and clock drift can introduce misalignment that accumulates over long horizons (Chen et al., 2025a; Liu et al., 2024). Practical remedies include timestamp normalization, learned shift predictors, cross-correlation to estimate lag, multi-scale encoders that align coarsely before refining, and anchoring on events shared across modalities (Xie et al., 2025; Liu et al., 2025c).

To address plotted-series bias and style overfitting, evaluations should incorporate render-swap controls and provide access to raw signals in addition to plots (Dong et al., 2024; Liu et al., 2025c). Outlook work should prioritize broader datasets and libraries with synchronized multimodal pairs and multilingual text, enabling more faithful evaluation of alignment under longer horizons and richer modalities (Liu et al., 2024; Liang et al., 2024).

### 7.3 Retrieval and Knowledge Grounding.

Grounding answers in external sources such as tables, knowledge bases, logs, and domain corpora is widely requested across tasks (Zhu et al., 2025; Zhang et al., 2025d; Yang et al., 2024). This line of work seeks to reduce hallucinations and improve domain specificity while maintaining efficiency (Hao et al., 2025; Gu et al., 2025; Zhang et al., 2025d). A central design choice is whether to retrieve at inference time or to pre-encode knowledge during training or adaptation. The decision depends on factors such as update frequency, domain shift, and memory or latency budgets (Tire et al., 2024; Bogahawatte et al., 2024; Liu et al., 2025h).

Time-aware retrieval is particularly promising. This involves building segment-level indexes over subsequence representations, aligning events and entities to timestamps, and retrieving windows that capture motifs or regimes rather than entire series (Tire et al., 2024; Yang et al., 2024). Tool-augmented retrieval provides further benefits by issuing structured queries such as SQL over tables, log filters, or simulator calls, and then feeding results back into the reasoning step with explicit citations (Zhu et al., 2025; Zhang et al., 2025d; Xu et al., 2025b).

When multiple candidates are retrieved, robustness improves through learned re-ranking with cross-modal checks and late fusion via evidence-weighted voting, rather than relying on a single source (Yao et al., 2025; Tire et al., 2024). Calibration also benefits from evidence-linked decoding, which constrains claims to cited spans, down-weights unsupported tokens, and abstains when retrieved evidence is weak or conflicting (Hao et al., 2025). For long data streams, these methods can be complemented by streaming retrieval with rolling caches and periodic re-indexing to maintain relevance without exceeding context limits (Tire et al., 2024; Yang et al., 2024).

The outlook is retrieval pipelines that are more time-aware, fault-tolerant, and auditable, supported by benchmarks that stress-test robustness under shift, incompleteness, and cost constraints (Hu et al., 2025).

### 7.4 Long Context, Memory, and Efficiency.

Scaling to longer histories with manageable latency and memory remains a recurring challenge (Tan et al., 2024; Zhou & Yu, 2025; Zhou et al., 2025a; Yu et al., 2024). Compression strategies include multi-resolution encoders, learnable downsampling of sequences or KV-caches, segment pooling or sketching, and value-aware sparsification that preserves extremes and change points (Chan et al., 2024; Zhou et al., 2025a; Yu et al., 2023). Streaming inference combines sliding windows with warm-start states and truncated backpropagation, with state handover across windows to avoid recomputing long prefixes (Kong et al., 2025a; Wang et al., 2025b). Stateful memory augments the model with episodic or event-indexed slots and explicit write/read policies so that long-range dependencies persist beyond the context window (Li et al., 2024a; Kong et al., 2025a).

Lightweight adaptation methods such as LoRA or IA3 adapters, prefix prompts, and linear probes for numeric channels help retain throughput while capturing domain specifics (Tao et al., 2024; Zhou et al., 2025a). Practical deployment further depends on compute-aware training and decoding, including mixed precision and quantization, block-sparse or chunked attention, early-exit or confidence-based halting, and speculative decoding to reduce latency (Hu et al., 2025; He et al., 2025; Chan et al., 2024; Li et al., 2025f).

Open problems include mitigating recency bias from limited windows, preserving temporal semantics under compression, and balancing efficiency with rare-event retention. For deployed streams, memory policies also become safety and privacy decisions: systems must decide which events to retain, summarize, retrieve, or delete while preserving enough provenance for later audits. Future progress will depend on benchmarks that jointly measure accuracy, latency, memory, energy, and retention risk, alongside protocols that capture streaming, drift-aware, and privacy-aware conditions (Liu et al., 2025g; Li et al., 2025f; Hu et al., 2025; Yu et al., 2024).

### 7.5 Agentic Control and Tool Use.

Many works envision systems that perceive streams, plan, call tools or simulators, and then verify and act, moving from passive prediction to closed-loop control (Zhang et al., 2024a; Yeh et al., 2025; Da et al., 2024; Cai et al., 2025). A central open problem is action selection under uncertainty and delayed feedback, where sparse rewards and partial observability make credit assignment brittle. Promising directions include uncertainty-aware planners, policy regularization, and counterfactual rollouts before committing to actions (Yu et al., 2024; Li et al., 2024a; Da et al., 2024).

Termination and rollback are often under-specified in deployed settings. Agents require explicit stop criteria, safe fallbacks, and recovery policies that bound risk during reversible and irreversible operations (Zhang et al., 2025c; Anonymous, 2025). Tool integration can be brittle: while domain solvers often improve correctness, they may increase latency and risk version drift. Recent approaches explore cost-aware tool selection, caching or batching of calls, and learned simulators that are periodically recalibrated to ground-truth solvers (Anonymous, 2025; Xiao et al., 2025b; Da et al., 2024).

Interface robustness is another recurring pain point, since schema or unit changes, simulator API evolution, and sampling inconsistencies often destabilize pipelines. Outlook work should add contract tests, unit and scale checks, and fault injection to agent evaluations (Anonymous, 2025; Da et al., 2024). Verification layers remain under-formalized: beyond heuristic checkers, agents need principled critics that test invariants, cross-validate with independent tools, and trigger abstention or rollback when evidence conflicts (Ye et al., 2025; Liu et al., 2025h; Jiang et al., 2025).

In deployed streaming settings, agentic time series systems should be evaluated as continuously operating services rather than isolated predictors. This requires bounded memory and retrieval policies, drift monitors, schema and API version tracking, cost-aware tool budgets, confidence thresholds, and fault-injection tests that cover stale context, missing windows, delayed labels, and failed tool calls. Such controls are especially important for branch-structured agents, where repeated retrieval, critique, and revision can amplify privacy leakage, automation bias, or temporal misalignment if intermediate evidence is not logged and checked.

Evaluation protocols have not yet kept pace with real-world practice. The community needs closed-loop benchmarks with standard tool APIs, rate limits, explicit costs, and safety budgets, reporting regret, constraint violations, and tool spend alongside accuracy (Anonymous, 2025; Cai et al., 2025; Da et al., 2024; Hu et al., 2025). Looking forward, human-in-the-loop governance will be critical, including audited logs, reproducible decision traces, and handoff protocols when confidence or safety falls below thresholds (Yeh et al., 2025).

### 7.6 Causal Inference and Decision Support.

Bridging descriptive explanations to causal conclusions, such as counterfactuals, treatment effects, and policies, remains a central goal for time series reasoning (Tan et al., 2024; Xu et al., 2025d; Liu et al., 2025c). Key open problems include identification under time-varying confounding, latent common causes, and feedback loops that arise in interactive or controlled settings (Xu et al., 2025d; Li et al., 2024b).

Benchmarks and simulators with known data-generating processes are needed, including longitudinal interventions, dynamic policies, and realistic constraints. Semi-synthetic designs that splice real series with scripted interventions can help establish ground truth (Da et al., 2024; Xu et al., 2025d; Liang et al., 2024). Methodologically, counterfactual forecasting and dynamic treatment learning should model interventions explicitly and test invariances implied by causal structure (Xu et al., 2025d; Li et al., 2024b).

Evaluations should link rationales to causal evidence through sufficiency and necessity checks, counterfactual consistency, and refutation tests such as placebos or sensitivity analyses (Liu et al., 2025h). Principled off-policy evaluation is required before deployment, combining importance-sampling and model-based estimators with calibrated uncertainty to report policy value, regret, and constraint violations (Yu et al., 2024; Cheng & Chin, 2024; Zhang et al., 2024a).

Heterogeneous effects and fairness under domain shift remain underexplored. Future work should report subgroup treatment effects with coverage guarantees and specify safety budgets for interventions (Belyaeva et al., 2023; Liu et al., 2024; Xu et al., 2025b). For high-stakes settings such as healthcare, finance, infrastructure, and operational decision support, broader impact depends on preventing unsafe intervention as much as improving prediction. Systems should expose calibrated uncertainty, privacy-aware retrieval and logging, explicit stop and rollback rules, and human handoff when evidence is stale, conflicting, or outside the intended deployment regime. A practical outlook is end-to-end pipelines that couple causal objectives with closed-loop evaluation, where policies are audited, costs are explicit, and rollback rules are triggered when uncertainty or shift exceeds thresholds (Da et al., 2024; Anonymous, 2025; Xu et al., 2025b).

## 8 Conclusion

Time series reasoning treats time as a first-class axis and integrates intermediate evidence into the answer itself. We organize the field by reasoning topology, distinguishing three main families: direct reasoning in a single step, linear chain reasoning with explicit intermediate steps, and branch-structured reasoning that explores, revises, and aggregates. Alongside topology, we consider the main objectives of the literature—traditional time series analysis, explanation and understanding, causal inference and decision making, and time series generation. Common techniques such as decomposition and verification, ensembling, tool use, knowledge access, multimodality, agent loops, and LLM adaptation regimes cut across these perspectives, offering a compact way to describe methods and their strengths and weaknesses.

Several themes emerge for design and evaluation. The choice of reasoning structure is central: moving from direct to chain to branch increases capacity for grounding, search, and self-correction, but also raises computational cost, variance, and reproducibility challenges. Evidence must remain visible and tightly linked to data, retrieved context, and tool outputs, with strict temporal alignment to keep narratives faithful to signals. Agents and tools can extend analysis into action, but they require clear stop rules, rollback plans, and cost-aware strategies. Evaluation should mirror deployment through shift-aware protocols, long horizons, and streaming settings, with checks that test whether rationales truly reflect the data. Cost and latency should be treated as design budgets, and lightweight adaptation often provides the right balance when domain specificity is needed.

Looking ahead, the field should pursue benchmarks that tie reasoning quality to utility, closed-loop testbeds that balance cost and risk, and streaming evaluations that capture long-horizon challenges. No single topology will dominate, as domains vary in constraints, costs, and tolerance for risk. What matters is deliberate structural choice, alignment with primary objectives, and evaluation that keeps evidence and faithfulness at the center. By advancing along these lines, time series reasoning can move from narrow accuracy toward broad reliability, enabling systems that not only analyze but also understand, explain, and act on dynamic worlds.

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

## A  Full Taxonomy Assignments

To keep the tables concise, we use abbreviations for reasoning topologies, objectives, tasks, and attribute tags. Table 2 lists these abbreviations. The complete taxonomy assignments of all curated **research papers** are then provided separately for each reasoning topology: Table 3 (direct reasoning), Table 4 (linear chain reasoning), and Table 5 (branch-structured reasoning). Each table reports the corresponding papers together with their primary objectives, specific tasks, and attribute tags. In addition, Table 6 reports curated **non-research papers**, which correspond to the resources surveyed in Section 6 (*Current Landscape and Resources*). Since attribute tags are not applicable, these works are presented only with their type and outlook. Together, these tables serve as a comprehensive reference for the taxonomy developed in the main text.

Table 2: Abbreviations and value definitions for table headers, primary objectives, task values, and attribute tags used in the curated research paper taxonomy tables (Table 3, Table 4, and Table 5). Non-research papers (Table 6) are listed without attribute tags.

| Full Name | Abbreviation / Values |
|---|---|
| *General Table Headers* | |
| Primary Objective | Prim. Obj. |
| *Attribute Tag Headers (8 total)* | |
| Task Decomposition | T-Dec |
| Verification and Critique | T-Ver |
| Ensemble Selection | T-Ens |
| Tool Use | T-Tool |
| Knowledge Access | T-Know |
| Multimodal Inputs | T-Multi |
| Agents | T-Agent |
| LLM Adaptation | T-Align |
| *Primary Objective Values* | |
| Traditional Time Series Analysis | Trad. TS Anal. |
| Explanation and Understanding | Expl. & Und. |
| Causal Inference and Decision Making | Causal Inf. |
| Time Series Generation | TS Gen. |
| *Task Values* | |
| Forecasting | Forc. |
| Classification | Class. |
| Anomaly Detection | Anom. Det. |
| Segmentation | Segm. |
| Multiple Tasks | Mult. Tasks |
| Temporal Question Answering | Temp. QA |
| Explanatory Diagnostics | Expl. Diagn. |
| Structure Discovery | Struct. Disc. |
| Autonomous Policy Learning | Auto. Policy |
| Advisory Decision Support | Adv. Dec. Supp. |
| Conditioned Synthesis | Cond. Synth. |
| *Attribute Tag Values* | |
| T-Dec, T-Ver, T-Ens, T-Tool, T-Know, T-Multi | ✓ = present,  empty = absent |
| T-Agent | 0 = no agent,  1 = single agent,  M = multiple agents |
| T-Align | P = Prompting,  S = Supervised fine-tuning |
| | R = Reinforcement/preference optimization,  H = Hybrid |

Table 3: Full taxonomy assignments of curated research papers with direct reasoning topology only, including primary objectives, tasks, and attribute tags.

| Method | Primary Obj. | Task | T-Dec | T-Ver | T-Ens | T-Tool | T-Know | T-Multi | T-Agent | T-Align |
|---|---|---|---|---|---|---|---|---|---|---|
| LLMTIME (Gruver et al., 2023) | Trad. TS Anal. | Forc. | | | ✓ | | | | 0 | P |
| CiK (Williams et al., 2025) | Trad. TS Anal. | Forc. | | | | | | ✓ | 0 | P |
| DP-GPT4MTS (Liu et al., 2025a) | Trad. TS Anal. | Forc. | | | | | | ✓ | 0 | S |
| TEMPO (Cao et al., 2024a) | Trad. TS Anal. | Forc. | ✓ | | | | | ✓ | 0 | S |
| NNCL-TLLM (Bogahawatte et al., 2024) | Trad. TS Anal. | Forc. | | | | | | | 0 | S |
| CMLLM (Zhu et al., 2025) | Trad. TS Anal. | Forc. | | | | | | | 0 | P |
| Hybrid-MMF (Kim et al., 2024) | Trad. TS Anal. | Forc. | | | | | | ✓ | 0 | S |
| Tang et al. (2025) | Trad. TS Anal. | Forc. | | | | | | | 0 | P |
| HiTime (Tao et al., 2024) | Trad. TS Anal. | Class. | | | | | | ✓ | 0 | S |
| HeLM (Belyaeva et al., 2023) | Trad. TS Anal. | Class. | | | | | | ✓ | 0 | S |
| FinSrag (Xiao et al., 2025a) | Trad. TS Anal. | Class. | | | | ✓ | ✓ | | 0 | S |
| Zhou & Yu (2025) | Trad. TS Anal. | Anom. Det. | ✓ | | | | | ✓ | 0 | P |
| MedTsLLM (Chan et al., 2024) | Trad. TS Anal. | Segm. | | | | | | ✓ | 0 | P |
| ChatTime (Wang et al., 2025a) | Trad. TS Anal. | Mult. Tasks | | | | | | ✓ | 0 | S |
| Chat-TS (Quinlan et al., 2025) | Expl. & Und. | Temp. QA | | | | | | ✓ | 0 | S |
| ChatTS (Xie et al., 2025) | Expl. & Und. | Temp. QA | | | | | | ✓ | 0 | S |
| ITFormer (wang et al., 2025) | Expl. & Und. | Temp. QA | | | | | | ✓ | 0 | P |
| Time-MQA (Kong et al., 2025a) | Expl. & Und. | Temp. QA | | | | | | ✓ | 0 | S |
| GEM (Lan et al., 2025) | Expl. & Und. | Expl. Diagn. | | | | | | ✓ | 0 | S |
| Time-RA (Yang et al., 2025b) | Expl. & Und. | Expl. Diagn. | ✓ | ✓ | ✓ | | | ✓ | 0 | S |
| Momentor (Qian et al., 2024) | Expl. & Und. | Expl. Diagn. | | | | | | ✓ | 0 | P |
| RealTCD (Li et al., 2024b) | Expl. & Und. | Struct. Disc. | | | | | | | 0 | P |
| GG-LLM (Graule & Isler, 2024) | Causal Inf. | Auto. Policy | | | | | | | 0 | P |

Table 4: Full taxonomy assignments of curated research papers with linear chain reasoning topology only, including primary objectives, tasks, and attribute tags.

| Method | Primary Obj. | Task | T-Dec | T-Ver | T-Ens | T-Tool | T-Know | T-Multi | T-Agent | T-Align |
|---|---|---|---|---|---|---|---|---|---|---|
| TimeReasoner (Wang et al., 2025c) | Trad. TS Anal. | Forc. | ✓ | ✓ | | | | ✓ | 0 | P |
| RAF (Tire et al., 2024) | Trad. TS Anal. | Forc. | | | | ✓ | ✓ | | 0 | S |
| TimeRAG (Yang et al., 2024) | Trad. TS Anal. | Forc. | | | | ✓ | ✓ | | 0 | P |
| Time-R1 (Luo et al., 2025) | Trad. TS Anal. | Forc. | ✓ | | | | | | 0 | H |
| Yu et al. (2023) | Trad. TS Anal. | Forc. | ✓ | | | ✓ | ✓ | ✓ | 0 | S |
| CAARL (Tajeuna et al., 2026) | Trad. TS Anal. | Forc. | ✓ | | | | | | 0 | P |
| TableTime (Wang et al., 2025d) | Trad. TS Anal. | Class. | ✓ | | ✓ | ✓ | | | 0 | P |
| VL-Time (Liu et al., 2025c) | Trad. TS Anal. | Class. | ✓ | | | | | ✓ | 0 | P |
| ZARA (Li et al., 2025e) | Trad. TS Anal. | Class. | ✓ | | | ✓ | ✓ | | M | P |
| TimeMaster (Zhang et al., 2025b) | Trad. TS Anal. | Class. | ✓ | ✓ | | ✓ | | ✓ | 0 | H |
| Chow et al. (2024) | Trad. TS Anal. | Class. | ✓ | | | | | ✓ | 0 | S |
| REALM (Zhu et al., 2024) | Trad. TS Anal. | Class. | ✓ | ✓ | | ✓ | ✓ | ✓ | 0 | P |
| VLM4TS (He et al., 2025) | Trad. TS Anal. | Anom. Det. | ✓ | ✓ | | | | ✓ | 0 | P |
| LLMAD (Liu et al., 2025e) | Trad. TS Anal. | Anom. Det. | ✓ | ✓ | | ✓ | ✓ | | 0 | P |
| Dong et al. (2024) | Trad. TS Anal. | Anom. Det. | ✓ | | | | | | 0 | S |
| SIGLLM (Alnegheimish et al., 2024) | Trad. TS Anal. | Anom. Det. | | | ✓ | | | | 0 | P |
| SLEP (Wang et al., 2026) | Trad. TS Anal. | Anom. Det. | | | ✓ | | | | 1 | P |
| LEMAD (Ji et al., 2025) | Trad. TS Anal. | Anom. Det. | ✓ | | | | | ✓ | M | P |
| LTM (Hao et al., 2025) | Trad. TS Anal. | Mult. Tasks | | | | ✓ | ✓ | ✓ | 0 | P |
| Ravuru et al. (2024) | Trad. TS Anal. | Mult. Tasks | | | | ✓ | ✓ | | M | H |
| Tan et al. (2025) | Expl. & Und. | Temp. QA | ✓ | | | | | ✓ | 0 | H |
| TG-LLM (Xiong et al., 2024) | Expl. & Und. | Temp. QA | ✓ | ✓ | | | | | 0 | S |
| TempoGPT (Zhang et al., 2025a) | Expl. & Und. | Expl. Diagn. | ✓ | | | | | ✓ | 0 | S |
| TSLM (Trabelsi et al., 2025) | Expl. & Und. | Expl. Diagn. | | | ✓ | ✓ | | ✓ | 0 | S |
| Xu et al. (2025a) | Expl. & Und. | Expl. Diagn. | ✓ | | | ✓ | ✓ | ✓ | M | P |
| FinAgent (Zhang et al., 2024a) | Causal Inf. | Auto. Policy | ✓ | ✓ | | ✓ | ✓ | ✓ | 1 | P |
| FINMEM (Li et al., 2024a) | Causal Inf. | Auto. Policy | ✓ | ✓ | | ✓ | ✓ | ✓ | 1 | P |
| Open-TI (Da et al., 2024) | Causal Inf. | Auto. Policy | ✓ | | | ✓ | | | M | P |
| SocioDojo (Cheng & Chin, 2024) | Causal Inf. | Adv. Dec. Supp. | ✓ | ✓ | | ✓ | ✓ | ✓ | M | P |
| GenG (Zhou et al., 2024) | TS Gen. | Cond. Synth. | ✓ | | | | | ✓ | 0 | S |
| Joshi (2025) | TS Gen. | Cond. Synth. | | ✓ | | ✓ | ✓ | | 0 | P |

Table 5: Full taxonomy assignments of curated research papers with branch-structured reasoning topology only, including primary objectives, tasks, and attribute tags.

| Method | Primary Obj. | Task | T-Dec | T-Ver | T-Ens | T-Tool | T-Know | T-Multi | T-Agent | T-Align |
|---|---|---|---|---|---|---|---|---|---|---|
| Zhang et al. (2025h) | Trad. TS Anal. | Forc. | ✓ | ✓ | ✓ | ✓ | ✓ | ✓ | M | S |
| NewsForecast (Wang et al., 2024) | Trad. TS Anal. | Forc. | ✓ | ✓ | | | ✓ | ✓ | M | S |
| CAPTime (Yao et al., 2025) | Trad. TS Anal. | Forc. | | | ✓ | | | ✓ | 0 | P |
| DCATS (Yeh et al., 2025) | Trad. TS Anal. | Forc. | ✓ | ✓ | | ✓ | ✓ | | 1 | P |
| CoLLM (Wang et al., 2025b) | Trad. TS Anal. | Forc. | | ✓ | ✓ | | | | 0 | S |
| TimeXL (Jiang et al., 2025) | Trad. TS Anal. | Forc. | ✓ | ✓ | ✓ | | | ✓ | M | P |
| CastFlow (Pan et al., 2026) | Trad. TS Anal. | Forc. | ✓ | ✓ | ✓ | ✓ | ✓ | | 1 | H |
| ReasonTSC (Zhou et al., 2025a) | Trad. TS Anal. | Class. | ✓ | ✓ | | ✓ | | | 0 | P |
| ColaCare (Wang et al., 2025) | Trad. TS Anal. | Class. | ✓ | ✓ | | ✓ | ✓ | ✓ | M | P |
| TimeCAP (Lee et al., 2025) | Trad. TS Anal. | Class. | ✓ | | ✓ | ✓ | ✓ | ✓ | M | P |
| AD-AGENT (Anonymous, 2025) | Trad. TS Anal. | Anom. Det. | ✓ | ✓ | | ✓ | ✓ | | M | P |
| ARGOS (Gu et al., 2025) | Trad. TS Anal. | Anom. Det. | ✓ | ✓ | ✓ | | | | 0 | P |
| TAMA (Zhuang et al., 2024) | Trad. TS Anal. | Anom. Det. | ✓ | ✓ | ✓ | | | ✓ | 0 | P |
| LLM-TSFD (Zhang et al., 2025d) | Trad. TS Anal. | Anom. Det. | ✓ | ✓ | | ✓ | ✓ | | 1 | P |
| TS-Reasoner (Ye et al., 2025) | Trad. TS Anal. | Mult. Tasks | ✓ | ✓ | | ✓ | ✓ | | 1 | P |
| MERIT (Zhou et al., 2025b) | Trad. TS Anal. | Mult. Tasks | ✓ | ✓ | | ✓ | ✓ | | M | P |
| TESSA (Lin et al., 2025) | Expl. & Und. | Expl. Diagn. | ✓ | ✓ | | | | ✓ | M | P |
| AgentFM (Zhang et al., 2025c) | Expl. & Und. | Expl. Diagn. | ✓ | | | ✓ | ✓ | ✓ | M | P |
| ElliottAgents (Chudziak & Wawer, 2024) | Expl. & Und. | Expl. Diagn. | ✓ | ✓ | | ✓ | ✓ | | M | P |
| Grammar of the Wave (Wan et al., 2026) | Expl. & Und. | Expl. Diagn. | ✓ | ✓ | | ✓ | ✓ | ✓ | M | P |
| Liu et al. (2025h) | Expl. & Und. | Struct. Disc. | ✓ | ✓ | ✓ | ✓ | | ✓ | 0 | P |
| FinArena (Xu et al., 2025b) | Causal Inf. | Auto. Policy | ✓ | ✓ | | ✓ | ✓ | ✓ | M | P |
| FINCON (Yu et al., 2024) | Causal Inf. | Auto. Policy | ✓ | ✓ | | ✓ | ✓ | ✓ | M | P |
| TradingAgents (Xiao et al., 2025b) | Causal Inf. | Auto. Policy | ✓ | ✓ | | ✓ | ✓ | ✓ | M | P |
| BRIDGE (Li et al., 2025b) | TS Gen. | Cond. Synth. | ✓ | ✓ | | ✓ | ✓ | ✓ | M | P |

Table 6: Curated non-research papers relevant to time series reasoning, including datasets, benchmarks, surveys, tutorials, and position/vision papers. These correspond to the resources discussed in Section 6 (Current Landscape and Resources).

| Paper | Type |
|---|---|
| Fons et al. (2024) | Reasoning-First Benchmarks |
| Potosnak et al. (2024) | Reasoning-First Benchmarks |
| Potosnak et al. (2025) | Reasoning-First Benchmarks |
| ReC4TS (Liu et al., 2025d) | Reasoning-First Benchmarks |
| MTBench (Chen et al., 2025a) | Reasoning-First Benchmarks |
| TSQA (Kong et al., 2025a) | Reasoning-First Benchmarks |
| PUB (Pawelec et al., 2024) | Reasoning-First Benchmarks |
| CiK (Williams et al., 2025) | Reasoning-First Benchmarks |
| TimeSeriesGym (Cai et al., 2025) | Reasoning-First Benchmarks |
| SocioDojo (Cheng & Chin, 2024) | Reasoning-First Benchmarks |
| Tan et al. (2025) | Reasoning-First Benchmarks |
| Temporal-Synced IATSF (Xu et al., 2025d) | Reasoning-First Benchmarks |
| EngineMT-QA (wang et al., 2025) | Reasoning-First Benchmarks |
| ECG-Grounding (Lan et al., 2025) | Reasoning-Ready Benchmarks |
| Zhuang et al. (2024) | Reasoning-Ready Benchmarks |
| GPT4MTS (Jia et al., 2024) | Reasoning-Ready Benchmarks |
| TimeTextCorpus (Kim et al., 2024) | Reasoning-Ready Benchmarks |
| STOCK23 (Xiao et al., 2025a) | Reasoning-Ready Benchmarks |
| TETS (Cao et al., 2024a) | Reasoning-Ready Benchmarks |
| DeepFund (Li et al., 2025a) | Reasoning-Ready Benchmarks |
| Moment-10M (Qian et al., 2024) | Reasoning-Ready Benchmarks |
| MoTime (Zhou et al., 2025c) | Reasoning-Ready Benchmarks |
| Time-IMM (Chang et al., 2025a) | Reasoning-Ready Benchmarks |
| Time-MMD (Liu et al., 2024) | Reasoning-Ready Benchmarks |
| TSFM-Bench (Li et al., 2025f) | Reasoning-Ready Benchmarks |
| VISUELLE (Skenderi et al., 2024) | Reasoning-Ready Benchmarks |
| RATs40K (Yang et al., 2025b) | Reasoning-Ready Benchmarks |
| TimerBed (Liu et al., 2025c) | General-Purpose Time Series Benchmarks |
| Tan et al. (2024) | General-Purpose Time Series Benchmarks |
| SymbolBench (Liu et al., 2025h) | General-Purpose Time Series Benchmarks |
| Zhou & Yu (2025) | General-Purpose Time Series Benchmarks |
| VISUALTIMEANOMALY (Xu et al., 2025c) | General-Purpose Time Series Benchmarks |
| TimeSeriesExam (Cai et al., 2024) | General-Purpose Time Series Benchmarks |
| TTGenerator (Dong et al., 2024) | General-Purpose Time Series Benchmarks |
| TSandLanguage (Merrill et al., 2024) | General-Purpose Time Series Benchmarks |
| TS Instruct (Quinlan et al., 2025) | General-Purpose Time Series Benchmarks |
| ChatTS (Xie et al., 2025) | General-Purpose Time Series Benchmarks |
| TS-Reasoner (Ye et al., 2025) | General-Purpose Time Series Benchmarks |
| FinBen (Xie et al., 2024) | General-Purpose Time Series Benchmarks |
| FinTSB (Hu et al., 2025) | General-Purpose Time Series Benchmarks |
| Liang et al. (2024) | Surveys and Tutorials |
| Su et al. (2024) | Surveys and Tutorials |
| Liu et al. (2025g) | Surveys and Tutorials |
| Miller et al. (2025) | Surveys and Tutorials |
| Liu et al. (2025b) | Surveys and Tutorials |
| Tan et al. (2024) | Position and Vision Papers |
| Zhang & Gilpin (2025) | Position and Vision Papers |
| Kong et al. (2025b) | Position and Vision Papers |
| Jin et al. (2024) | Position and Vision Papers |

