# OpenReview forum: "A Survey of Reasoning and Agentic Systems in Time Series with Large Language Models"
_TMLR — Accepted by TMLR_

### Review · Reviewer_hW3H · 2026-03-03

**Summary Of Contributions:**

This paper presents a systematic survey of time series reasoning with large language models (LLMs) and agentic systems. Rather than organizing prior work by application domain or modeling technique, the authors introduce a structured taxonomy centered on reasoning topology, which categorizes methods into three mutually exclusive execution structures: Direct reasoning, Linear chain reasoning, and Branch-structured reasoning. To further characterize the methods, the paper introduces a set of lightweight attribute tags that capture control-flow operators (e.g., decomposition, critique, ensembling), execution actors (e.g., tools, agents), information sources (e.g., multimodality, knowledge access), and LLM adaptation regimes (e.g., prompting, SFT, RL-based alignment, hybrid).

Strengths
- Clear and principled taxonomy
- Comprehensive coverage
- System-oriented perspective

Weaknesses
- Structural rather than semantic formalization. The taxonomy focuses on execution topology but does not formalize the correctness, faithfulness, or soundness of reasoning.
- Evaluation discussion seems high-level

**Audience:**

Yes

**Audience Explanation:**

This paper provides: a structural taxonomy of reasoning execution (direct, linear, branch), a framework for comparing reasoning architectures that aligns well with ongoing work on structured inference and LLM orchestration.

**Broader Impact Concerns:**

None.

**Claims And Evidence:**

No

**Claims Explanation:**

The paper makes a meaningful conceptual contribution by organizing the emerging TSR field around reasoning structure and objectives.

**Requested Changes:**

- Strengthen the Discussion of Evaluation Gaps
While the paper emphasizes reliability, trace visibility, and grounding, the evaluation section remains high-level. The authors should:
  + Provide more concrete proposals or exemplars of reasoning-quality metrics,
  + Clarify how reasoning topology affects measurable robustness,
  + Discuss reproducibility challenges in branch-structured systems.

- Provide a Compact Summary Table of Cross-Topology Trends

A consolidated table comparing strengths/weaknesses across topologies (e.g., interpretability, robustness, cost, reproducibility) would enhance accessibility.

---

> ### Author Response · Authors · 2026-05-10
>
> Thank you for the thoughtful review and for highlighting the value of the taxonomy and system-oriented perspective. We agree with your main concern that a structural taxonomy should be paired with more concrete guidance on evaluation, robustness, and reproducibility. We revised the manuscript accordingly.
>
> First, we clarified the scope and limitation of the taxonomy in Section 2.2. The revised text now states that topology labels are assigned from the observable execution trace, while primary objectives are assigned from the dominant evaluation target or main claimed contribution. We also explicitly note that these labels organize execution structure and survey analysis, but do not by themselves certify correctness, faithfulness, causal validity, or robustness. Those properties must be evaluated separately.
>
> Second, we substantially expanded Section 7.1, "Evaluation and Benchmarking." The revised section separates evaluation into three layers: output-level metrics, reasoning-level faithfulness and grounding metrics, and topology-level robustness and reproducibility metrics. We now give concrete examples, including sufficiency and necessity of cited evidence, explanation-data agreement, temporal localization accuracy, rationale/numeric consistency, horizon shift, domain shift, missingness, perturbation tests, prompt/seed sensitivity, latency, tool spend, branch-budget sensitivity, and stopping-rule sensitivity.
>
> Third, we added Table tab:topology_evaluation, a compact cross-topology table in Section 7.1 comparing direct, linear-chain, and branch-structured reasoning across inspectable evidence, robustness pattern, cost/latency, and reproducibility requirements. This directly addresses your request for a consolidated summary of topology-level trends. We also added branch-specific reproducibility guidance: evaluations should report branch budgets, pruning rules, stopping criteria, aggregation policies, random seeds, model/tool versions, and whether full branch traces are released.
>
> Finally, we made the main limitation explicit: greater topological complexity does not automatically imply greater correctness. Topology changes what evidence can be inspected, where errors can be checked, and whether alternatives can be revised or aggregated, but correctness still depends on task-specific metrics, stress tests, and reproducible execution logs.

---

### Review · Reviewer_dr44 · 2026-03-05

**Summary Of Contributions:**

This paper presents a survey of time series reasoning (TSR) with large language models and agentic systems. The work proposes a conceptual framework that organizes the literature along three reasoning topologies: (1) direct reasoning, (2) linear chain reasoning, and (3) branch-structured reasoning. These reasoning structures are further connected to several primary objectives such as traditional time series analysis, explanation and interpretation, causal reasoning and decision making, and time series generation.

Strengths
1. The intersection of large language models, reasoning paradigms, and time series analysis is an emerging research direction. The survey attempts to summarize recent developments and identify promising research opportunities.
2. The survey organizes prior work along several dimensions including reasoning structure, application objectives, and system attributes. This helps structure the rapidly growing literature.
3. The paper outlines several open challenges such as evaluation protocols, multimodal reasoning, retrieval grounding, and agent-based time series analysis.

Weaknesses
1. The paper mainly proposes conceptual taxonomies and classifications but does not provide empirical analyses or case studies to validate the usefulness of the proposed framework. More in-depth comparisons between representative methods would significantly improve the value of the survey.
2. Although cost and latency are mentioned, the survey could further analyze scalability, reliability, and real-world deployment challenges, especially for agentic systems interacting with streaming time series.
3. The notion of TSR appears broad and sometimes overlaps with standard applications of LLMs to time series tasks. A clearer and more rigorous definition would improve conceptual clarity.
4. The submission appears to contain identifying information in the provided materials, which may conflict with the double-blind reviewing policy. The authors should ensure that all materials (including supplementary files or external links) are properly anonymized to comply with the review guidelines.

**Audience:**

Yes

**Audience Explanation:**

The topic of applying large language models and reasoning frameworks to time series analysis is timely and relevant to several communities, including machine learning and time series research. A structured overview of this emerging area could be useful for researchers entering the field.

**Broader Impact Concerns:**

The work is a survey and conceptual framework for organizing research on time series reasoning with large language models, and it does not introduce new algorithms or systems that would immediately raise ethical or societal concerns. Therefore, no significant broader impact issues are apparent. Nonetheless, future work in this area may need to consider reliability, transparency, and potential misuse of automated decision-making systems based on time series data.

**Claims And Evidence:**

No

**Claims Explanation:**

The paper mainly proposes a conceptual taxonomy and literature categorization, but does not provide empirical analyses or case studies to demonstrate the usefulness of the proposed framework. As a result, the claims are supported primarily by qualitative discussion rather than convincing evidence. In addition, some materials appear to contain identifying information, which may conflict with the double-blind review requirement.

**Requested Changes:**

See above.

---

> ### Author Response · Authors · 2026-05-10
>
> Thank you for the careful review and for identifying several areas where the survey needed more rigor and practical grounding. We agree that the original version could more clearly distinguish time series reasoning from generic LLM-for-time-series applications, document the survey methodology, and synthesize the taxonomy's utility beyond cataloging papers.
>
> To address the definition concern, we revised Section 2.1 to sharpen what we mean by time series reasoning. We now distinguish TSR from generic LLM-for-time-series applications by requiring that temporal data shape an explicit reasoning procedure, evidence trace, tool interaction, or agentic workflow, rather than merely serving as input to a black-box predictor. This boundary is also reflected in the labeling protocol added in Section 2.2.
>
> To address survey rigor, we added Section 2.2, "Survey Corpus Construction." This subsection states that the corpus covers work available up to April 2026 and describes how papers were collected through keyword searches, source and venue scanning, arXiv/OpenReview searches, and follow-up through references and closely related work. It also lists the search themes and explicit inclusion/exclusion criteria. Included papers must place time series data at the center of the problem setting and involve LLMs, multimodal LLMs, explicit reasoning traces, tool use, agentic workflows, or related reasoning-oriented mechanisms. We exclude conventional time series papers without an LLM or reasoning component, generic LLM papers without a substantive temporal-data setting, duplicate versions, and papers where time series are only incidental.
>
> We also strengthened the analytical synthesis without changing the paper into a new benchmark study. Since this is a survey, we do not claim new empirical experiments. Instead, we added representative cross-topology comparisons in Section 7.1 for anomaly detection, context-aware forecasting, and decision support. These comparisons show how the taxonomy functions as a diagnostic lens: it identifies where evidence enters, where errors can be checked, and what evaluation controls are needed. We also explicitly caution that current cross-paper evidence remains weak when methods are evaluated on different datasets, shifts, prompt budgets, and tool stacks.
>
> For scalability, reliability, and deployment, we expanded Sections 7.4-7.6: "Long Context, Memory, and Efficiency," "Agentic Control and Tool Use," and "Causal Inference and Decision Support." The revised manuscript now discusses bounded memory and retrieval policies, drift monitors, schema and API version tracking, cost-aware tool budgets, confidence thresholds, fault-injection tests, stop and rollback rules, and human handoff. These additions are especially focused on streaming and agentic TSR systems, which should be evaluated as continuously operating services rather than isolated predictors.
>
> Finally, regarding anonymization, thank you for flagging the issue. The problem was an accidental link in the anonymous repository README to a non-anonymous arXiv preprint. We deleted that repository and created a strictly anonymized replacement repository. The corrected anonymous link was shared with reviewers by the Action Editor on March 29, 2026. In the revision, we updated the manuscript's repository URL to the corrected anonymous link and rechecked the review-facing materials.

---

### Review · Reviewer_VRuL · 2026-04-26

**Summary Of Contributions:**

This paper presents a survey of time series reasoning (TSR) with LLMs and agentic systems, and organizes the area using a two-axis taxonomy: reasoning topology (direct, linear-chain, branch-structured) and primary objective (traditional time-series analysis, explanation/understanding, causal inference & decision making, and time-series generation), with additional non-exclusive attribute tags for control flow, tools/agents, knowledge access, multimodality, and adaptation regime. It also curates datasets/benchmarks, surveys/position papers, controversies/counter-evidence, and open problems, and states that the survey includes a reproducible labeling of over one hundred papers plus an accompanying repository.

**Additional Comments:**

Overall, I found this to be a strong and timely survey. The topic is important, the taxonomy is useful, and the paper is likely to serve as a valuable entry point for researchers working on LLMs, reasoning, agents, and time-series modeling. My main reservation is not about relevance or quality, but about survey rigor: the paper would be considerably stronger if the literature-selection and annotation methodology were documented more explicitly. With that improvement, I would view this as a solid contribution for TMLR.

**Audience:**

Yes

**Audience Explanation:**

The paper targets a timely intersection of LLMs, agentic systems, and time-series reasoning, which is likely to interest multiple parts of the TMLR audience: researchers in LLM reasoning/agents, multimodal ML, time-series modeling, and evaluation. Its taxonomy of direct, linear-chain, and branch-structured reasoning, combined with objective-based categorization and resource curation, gives readers a useful framework for organizing a fast-growing area and identifying open problems in benchmarking, grounding, efficiency, and decision support. Even for readers who do not work directly on time series, the paper’s discussion of reasoning structure, tool use, and evaluation under dynamic settings is broadly relevant.

**Broader Impact Concerns:**

I do not see a major unaddressed ethical issue that would block publication. However, because the survey explicitly covers applications in healthcare, finance, and decision support, and discusses agentic control and intervention-oriented settings, I would encourage the authors to add a short broader-impact discussion emphasizing risks of automation bias, temporal misalignment, unsafe intervention, privacy leakage from retrieved data, and the need for rollback/handoff mechanisms in high-stakes deployments. The paper already gestures in this direction in its open-problem and conclusion sections, so this would mainly make those concerns more explicit.

**Claims And Evidence:**

Yes

**Claims Explanation:**

For a survey paper, the main claims are conceptual and organizational rather than experimental, and I think the paper generally supports them well. The taxonomy is clearly defined, visually motivated, and consistently applied through the paper; the resource section and appendix tables help substantiate the breadth of coverage; and the discussion of controversies and open problems shows that the authors are not merely listing papers but attempting to distill a field-level perspective. The claim that the survey provides a unified taxonomy, a curated view of datasets/benchmarks/resources, and a forward-looking agenda is therefore broadly supported by the manuscript.

**Requested Changes:**

1. Clarify the survey methodology.
Please add a more explicit description of how papers were collected, what time period and venues were covered, what inclusion/exclusion criteria were applied, and how the final curated set was formed.
2. Strengthen the labeling protocol.
Since a central contribution is the taxonomy and “reproducible labeling pipeline,” please describe how taxonomy assignments were made in practice, how ambiguous/hybrid papers were handled, and whether there was any cross-checking or inter-annotator agreement.
3. Discuss taxonomy limitations more directly.
The mutually exclusive assignment of one topology and one objective is clean, but some methods likely span multiple objectives or mix chain and branch behaviors. Please include a short “threats to validity / limitations of the taxonomy” discussion.
4. Increase analytical synthesis.
In several subsections, the survey reads as a structured catalog. Please add a stronger cross-paper synthesis: e.g., what trends appear to generalize across domains, which topology choices seem robust under shift, and where current evidence is still weak.

---

> ### Author Response · Authors · 2026-05-10
>
> Thank you for the positive and constructive review. We appreciate your assessment that the taxonomy is useful and that the paper can serve as a timely entry point for researchers working on LLMs, reasoning, agents, and time-series modeling. We agree that the main area needing improvement was survey rigor: the corpus construction, labeling protocol, taxonomy limitations, and cross-paper synthesis needed to be documented more explicitly.
>
> In response, we added Section 2.2, "Survey Corpus Construction." The revised section states that the survey covers work available up to April 2026 and explains the collection process, including keyword searches, source and venue scanning, arXiv/OpenReview searches, and follow-up through references and related work. It also lists search themes such as time series reasoning, LLMs for time series, forecasting, anomaly detection, temporal question answering, multimodal time series, tool use, agents, and decision support. We added explicit inclusion and exclusion criteria and clarified how duplicate or multiple versions of the same work were handled.
>
> We also strengthened the labeling protocol in Section 2.2. The revised text explains that topology labels are assigned from the observable execution trace; primary objectives from the dominant evaluation target or main claimed contribution; task labels from the concrete problem setting; and attribute tags only when explicitly evidenced. For hybrid cases, topology and objective remain mutually exclusive for readability and table consistency, while non-exclusive attribute tags record secondary behaviors such as decomposition, verification, tool use, knowledge access, multimodality, agents, and adaptation.
>
> We did not measure formal inter-annotator agreement on the labeled corpus. Instead, we documented deterministic assignment rules in Section 2.2 -- topology from the observable trace, primary objective from the dominant evaluation target, task labels from the concrete problem setting, and attribute tags only when explicitly evidenced -- so that labels can be re-derived from the rules and cited sources. We agree this is a limitation and have noted that the taxonomy organizes execution structure rather than certifying correctness, faithfulness, causal validity, or robustness.
>
> To address taxonomy limitations, we added an explicit limitation statement: the taxonomy describes execution structure and survey organization, but does not itself certify correctness, faithfulness, causal validity, or robustness. We also expanded Section 7.1 to show how those properties should be evaluated separately through output-level, reasoning-level, and topology-level metrics.
>
> To increase analytical synthesis, we revised the Introduction around Figure 1 and added a representative comparison paragraph in Section 7.1. The Introduction now explains that topology changes what evidence is visible, where errors can be checked, and whether competing hypotheses can be revised or aggregated. Section 7.1 then compares topology choices across anomaly detection, context-aware forecasting, and decision support. We frame the taxonomy as a diagnostic lens rather than a performance ranking, and explicitly caution that topology-level robustness claims should be stress-tested because current cross-paper evidence often differs in datasets, shifts, prompt budgets, and tool stacks.
>
> Finally, we added a more explicit broader-impact and deployment discussion in Sections 7.4-7.6: "Long Context, Memory, and Efficiency," "Agentic Control and Tool Use," and "Causal Inference and Decision Support." The revised text discusses automation bias, temporal misalignment, unsafe intervention, privacy leakage from repeated retrieval/logging, stop and rollback rules, calibrated uncertainty, and human handoff when evidence is stale, conflicting, or outside the intended deployment regime.

---

### Author Response · Authors · 2026-05-10

Dear Action Editor and Reviewers,

Thank you for your constructive feedback. We have submitted a revised version of the manuscript and have responded to all reviewer comments.

In this revision, we strengthened the survey methodology, clarified the labeling protocol and taxonomy limitations, expanded the evaluation and robustness discussion, added a compact cross-topology comparison table, improved analytical synthesis across representative task families, expanded the deployment and broader-impact discussion, and updated the manuscript to use the corrected anonymous repository link.

We appreciate your time and consideration.

---

### Author Response · Authors · 2026-06-10
**Camera-Ready Submission Completed**

Dear Action Editor Zhou and Reviewers,

We have uploaded the camera-ready version of our manuscript, incorporating the final de-anonymization, author metadata, and repository links.

We would like to take this opportunity to thank the Action Editor and the reviewers once again for your time, constructive feedback, and support throughout this review cycle.

Best regards,
Authors of Submission 7682

---

### Decision · Action_Editor_4KZL · 2026-06-09

**Recommendation:** Accept as is

**Audience:**

Yes

**Audience Explanation:**

Yes. The paper covers a timely and important topic, and its taxonomy and survey of recent developments would be of broad interest to the TMLR community

**Claims And Evidence:**

Yes

**Claims Explanation:**

Yes. The claims are generally well supported by a comprehensive literature review, a clearly defined taxonomy, and extensive coverage of existing works. The paper provides convincing evidence through systematic categorization, comparative analysis, and discussion of datasets, benchmarks, and open challenges.